# The 2024 Foundation Model Transparency Index

**Rishi Bommasani***  
*Stanford University*
nlprishi@stanford.edu

**Kevin Klyman***  
*Stanford University*
kklyman@stanford.edu

**Sayash Kapoor**  
*Princeton University*
sayashk@princeton.edu

**Shayne Longpre**  
*Massachusetts Institute of Technology*
slongpre@media.mit.edu

**Betty Xiong**  
*Stanford University*
xiongb@stanford.edu

**Nestor Maslej**  
*Stanford University*
nmaslej@stanford.edu

**Percy Liang**  
*Stanford University*
pliang@cs.stanford.edu

**Reviewed on OpenReview:** *https://openreview.net/forum?id=38cwP8xVxD*

## Abstract

Foundation models are increasingly consequential yet extremely opaque. To characterize the status quo, the Foundation Model Transparency Index was launched in October 2023 to measure the transparency of leading foundation model developers. The October 2023 Index (v1.0) assessed 10 major foundation model developers (e.g. OpenAI, Google) on 100 transparency indicators (e.g. does the developer disclose the wages it pays for data labor?). At the time, developers publicly disclosed very limited information with the average score being 37 out of 100. To understand how the status quo has changed, we conduct a follow-up study (v1.1) after 6 months: we score 14 developers against the same 100 indicators. While in v1.0 we searched for publicly available information, in v1.1 developers submit reports on the 100 transparency indicators, potentially including information that was not previously public. We find that developers now score 58 out of 100 on average, a 21 point improvement over v1.0. Much of this increase is driven by developers disclosing information during the v1.1 process: on average, developers disclosed information related to 16.6 indicators that was not previously public. We observe regions of sustained (i.e. across v1.0 and v1.1) and systemic (i.e. across most or all developers) opacity such as on copyright status, data access, data labor, and downstream impact. We publish *transparency reports* for each developer that consolidate information disclosures: these reports are based on the information disclosed to us via developers. Our findings demonstrate that transparency can be improved in this nascent ecosystem, the Foundation Model Transparency Index likely contributes to these improvements, and policymakers should consider interventions in areas where transparency has not improved.

# 1 Introduction

Foundation models are the epicenter of artificial intelligence (AI) as AI begins to shape how the economy and society function (Bommasani et al., 2021). For such a high-impact technology, transparency is vital to facilitate accountability, competition, and collective understanding. As an illustrative example, the current lack of transparency regarding the data used to build foundation models makes it difficult to assess what copyrighted information is used to train foundation models. Governments around the world are intervening to increase transparency: for example, the EU AI Act and the US's proposed AI Foundation Model Transparency Act take major strides by mandating a number of disclosure requirements (Bommasani et al., 2024, Appendix A).

To characterize the transparency of the foundation model ecosystem, Bommasani et al. (2023a) introduced the Foundation Model Transparency Index (FMTI). Launched in October 2023, the first iteration of the index (FMTI v1.0) scored 10 major foundation developers (e.g. OpenAI, Google, Meta) based on publicly available information regarding 100 transparency indicators. These 100 indicators span matters such as the data, labor, and compute used to build models; the capabilities, limitations, and risks associated with models; and the distribution of models as well as the impact of their use. FMTI v1.0 established that, in the status quo, the foundation model ecosystem was opaque: the average score was 37 points out of 100. Yet FMTI v1.0 also identified heterogeneity in public disclosures: while the top score was a 54, for 82 indicators at least one developer scored a point.

To understand how the landscape has evolved in the last 6 months, we conduct a follow-up study (FMTI v1.1).[1] To enable direct comparison, we retain the 100 transparency indicators and the associated threshold for awarding a point from FMTI v1.0. However, instead of searching for public information as was done in FMTI v1.0, we request that developers report the relevant information for each indicator. We implemented this change for three reasons: (i) **completeness:** we obviate the concern that information was missed when searching the Internet; (ii) **clarity:** we reduce uncertainty by having developers affirmatively disclose information; and (iii) **scalability:** we remove the effort required for researchers to conduct an open-ended search for decentralized public information.

We contacted 19 foundation model developers, and 14 provided reports related to the 100 transparency indicators (Adept, AI21 Labs, Aleph Alpha, Amazon, Anthropic, BigCode/Hugging Face/ServiceNow, Google, IBM, Meta, Microsoft, Mistral, OpenAI, Stability AI, Writer).[2] Given each developer's initial report, we provided scores based on whether each disclosure satisfied the associated indicator. Developers responded to these initial scores, engaging in dialogue via email and virtual meetings, and clarifying matters in many cases. Following this iterative process, for each developer we publish a *transparency report* that consolidates the information it discloses. These reports contain new information, which developers had not disclosed publicly prior to the start of FMTI v1.1. On average, developers disclosed information related to 16.6 indicators that was not previously public.

The FMTI v1.1 results demonstrate ample room for improvement, as well as tangible improvements in transparency over half a year. On average, developers disclose information that satisfies 58 of the 100 transparency indicators. Developers are least transparent with respect to the upstream resources required to build their foundation models, scoring 46%, in comparison to 65% on downstream indicators and 61% on model-related indicators. Developers can become more transparent by drawing on the transparency practices of other developers—at least one developer scores a point on 96 of the 100 indicators, and multiple developers score a point on 89 indicators.

We find that developers' scores improved significantly over FMTI v1.0, with a 21 point improvement in the mean overall score. Scores improved across every domain, with upstream, model, and downstream scores improving by 6–7 points. Each of the 8 developers that were evaluated in both v1.0 and v1.1 improved their scores, with an average increase of 19 points. While some developers disclosed substantially more (e.g. AI21

---

[1] https://crfm.stanford.edu/fmti

[2] The change from FMTI v1.0 is the inclusion of 6 developers (Adept, Aleph Alpha, IBM, Microsoft, Mistral, Writer) and the exclusion of 2 developers (Cohere, Inflection).

Labs's score increased by 50 points), others made fairly marginal changes (e.g. OpenAI's score increased by just 1 point).

These findings affirm that greater transparency is feasible in the foundation model ecosystem. Further, they suggest that the Foundation Model Transparency Index in tandem with other interventions drives improvements in transparency. However, given that there has been very little progress on specific indicators (e.g. external data access, mitigations evaluations), we encourage policymakers to assess what level of minimum transparency is needed and to pursue policy interventions accordingly. We publish the transparency reports for all developers to enable further research.[3]

## 2 Background

To contextualize our effort, we describe FMTI v1.0 and prior work on multi-iteration indices.

### 2.1 The Foundation Model Transparency Index

Bommasani et al. (2023a) launched the Foundation Model Transparency Index in October 2023. To conceptualize transparency for foundation models, they introduced a hierarchical taxonomy aligned with the foundation model supply chain (Bommasani et al., 2023b). This taxonomy featured three top-level *domains*: the *upstream* resources involved in developing a foundation model, the foundation *model* itself and its properties, and the *downstream* use of the foundation model. These domains aggregate 23 subdomains (e.g. the upstream domain contains data, labor, data access, and compute as subdomains) and 100 binary transparency *indicators*. Using public information identified through a systematic search protocol, FMTI v1.0 scored 10 companies (AI21 Labs, Amazon, Anthropic, Cohere, Google, BigScience/Hugging Face, Inflection, Meta, OpenAI, Stability AI) from 0–100 on the 100 indicators. Companies were sent initial scores and allowed to contest them before FMTI v1.0 was released.

FMTI v1.0 showed a pervasive lack of transparency in the foundation model ecosystem, with the highest-performing developer scoring just 54 out of 100. Developers disclosed very little information about the labor or compute used to build their foundation models (scoring just 17% on these subdomains), or their real world impact (scoring 11% on this subdomain). Open foundation model developers—which refers to developers who released their flagship foundation model openly (i.e. with widely available model weights; Kapoor et al., 2024)—outperformed closed developers by a wide margin: all three developers with open flagship foundation models were among the top four scoring developers. Several companies disclosed almost no information about their flagship foundation models, with three companies scoring 25% or less.

### 2.2 Indices over time

A central objective of an index is to track a concept over time to characterize changes. In doing so, many notable indices have evolved over time to reflect changing circumstances and priorities. For instance, the Human Development Index (HDI) was changed multiple times in the 1990s and 2000s, largely in response to academic criticism (Klasen, 2018; Stanton, 2007). Despite these changes, which complicate direct comparisons across index iterations, the HDI remains one of the most trustworthy and popular indices for human development.

As an index is conducted repeatedly, and the world changes as measured by the index, a natural question is how the index contributes to this change. Attributing why corporate behavior (such as disclosure practices) changes is notoriously difficult. Companies generally do not reveal why they make changes and changes generally reflect a confluence of multiple factors. Kogen (2022) provides a unique demonstration of an index's impact, analyzing the 2018 Ranking Digital Rights Index (RDR), which ranked the freedom of expression and privacy policies of 26 of the world's largest ICT companies. By reviewing internal RDR documents and interviewing relevant stakeholders (e.g. representatives from 11 companies and 14 civil society groups), Kogen concluded that RDR had clear influence and that indexes can be useful resources for social movements. In the case of FMTI, we expect that the Index brings attention to the disclosure practices of companies,

---

[3]https://www.github.com/stanford-crfm/fmti

making it easier for media, policymakers, investors, customers and the public to apply additional pressure that engenders greater transparency. Additionally, the Index provides clarity to companies by setting concrete targets and empowers employees within companies to push for greater transparency.

To reason about an index's impact over time, we also draw inspiration from Raji & Buolamwini (2019). In 2017, Buolamwini & Gebru (2018) demonstrated significant performance disparities across demographic groups in 3 face recognition systems (from IBM, Microsoft, and Megvii). A year later, Raji & Buolamwini (2019) audited five systems (IBM, Microsoft, Megvii, Amazon, Kairos): they found that the original 3 systems had reduced the performance disparities considerably, whereas the 2 new systems in 2018 showed large disparities comparable to those seen in the 3 systems from 2017. In §4.2, we similarly explore how foundation model developers fare in FMTI v1.1 when stratified by whether they were assessed in FMTI v1.0.

## 3 Methods

FMTI v1.1 involves four steps: indicator selection, developer selection, information gathering, and scoring. We describe these steps and how they relate to their implementation in FMTI v1.0 below.

### 3.1 Indicator selection

To concretize transparency, we use the 100 indicators from FMTI v1.0: Bommasani et al. (2023a) defined these indicators based on the literature on AI transparency. The 100 indicators are listed in Figure 1.[4] The indicators are organized hierarchically into 3 domains (upstream, model, downstream) and several subdomains therein.

32 upstream indicators address the resources involved in foundation model development. Primarily, this relates to transparency around training data, labor practices, computational costs, and code/technical decisions. For example, the compute subdomain covers matters like the amount of hardware used to train a model, the owner of that hardware, the associated computational cost and training duration, and resulting energy and environmental impacts.

33 model indicators address the model iteself. Primarily, this relates to transparency around basic model properties, model access, capabilities, risks, and mitigations. For example, the risk subdomain covers whether risks are enumerated and are legible to laypersons, whether risks are evaluated pre-deployment for both unintentional (e.g. bias) and malicious (e.g. disinformation) type risks, and whether external parties evaluate risk.

35 downstream indicators address the distribution and usage of models. Primarily, this relates to distribution practices, usage policies, privacy protections, and downstream impact in society and the economy. For example, the distribution subdomain covers how release decisions are made by the developer, where the model is distributed, and whether it is integrated into other products and services by the developer.

Overall, these indicators are the byproduct of a wealth of research in these different subdomains spanning labor (Gray & Suri, 2019a; Crawford, 2021; Hao & Seetharaman, 2023), data (Bender & Friedman, 2018; Gebru et al., 2018; Longpre et al., 2023b;a), compute (Lacoste et al., 2019; Schwartz et al., 2020; Patterson et al., 2021; Luccioni & Hernández-García, 2023), evaluation (Liang et al., 2023), safety (Cammarota et al., 2020; Longpre et al., 2024a), privacy (EU, 2016; Brown et al., 2022; Vipra & Myers West, 2023; Winograd, 2023), policies (Kumar et al., 2022; Weidinger et al., 2021; Brundage et al., 2020), and impact (Tabassi, 2023; Weidinger et al., 2023).

### 3.2 Developer selection

In FMTI v1.0, Bommasani et al. (2023a) selected 10 foundation model developers: all 10 were companies developing salient foundation models with consideration given for diversity (e.g. type of company, type of foundation model). Further, for each foundation model developer, Bommasani et al. (2023a) designated a *flagship* foundation model that was used as the basis for scoring the developer. In FMTI v1.1, we require

---

[4]For full definitions, see `https://www.github.com/stanford-crfm/fmti`.

**2023 Foundation Model Transparency Index Indicators**

| Upstream | Model | Downstream |
|---|---|---|
| Data size | Input modality | Release decision-making |
| Data sources | Output modality | Release process |
| Data creators | Model components | Distribution channels |
| Data source selection | Model size | Products and services |
| Data curation | Model architecture | Detection of machine-generated content |
| Data augmentation | Centralized model documentation | Model License |
| Harmful data filtration | External model access protocol | Terms of service |
| Copyrighted data | Blackbox external model access | Permitted and prohibited users |
| Data license | Full external model access | Permitted, restricted, and prohibited uses |
| Personal information in data | Capabilities description | Usage policy enforcement |
| Use of human labor | Capabilities demonstration | Justification for enforcement action |
| Employment of data laborers | Evaluation of capabilities | Usage policy violation appeals mechanism |
| Geographic distribution of data laborers | External reproducibility of capabilities evaluation | Permitted, restricted, and prohibited model behaviors |
| Wages | Third party capabilities evaluation | Model behavior policy enforcement |
| Instructions for creating data | Limitations description | Interoperability of usage and model behavior policies |
| Labor protections | Limitations demonstration | User interaction with AI system |
| Third party partners | Third party evaluation of limitations | Usage disclaimers |
| Queryable external data access | Risks description | User data protection policy |
| Direct external data access | Risks demonstration | Permitted and prohibited use of user data |
| Compute usage | Unintentional harm evaluation | Usage data access protocol |
| Development duration | External reproducibility of unintentional harm evaluation | Versioning protocol |
| Compute hardware | Intentional harm evaluation | Change log |
| Hardware owner | External reproducibility of intentional harm evaluation | Deprecation policy |
| Energy usage | Third party risks evaluation | Feedback mechanism |
| Carbon emissions | Mitigations description | Feedback summary |
| Broader environmental impact | Mitigations demonstration | Government inquiries |
| Model stages | Mitigations evaluation | Monitoring mechanism |
| Model objectives | External reproducibility of mitigations evaluation | Downstream applications |
| Core frameworks | Third party mitigations evaluation | Affected market sectors |
| Additional dependencies | Trustworthiness evaluation | Affected individuals |
| Mitigations for privacy | External reproducibility of trustworthiness evaluation | Usage reports |
| Mitigations for copyright | Inference duration evaluation | Geographic statistics |
| | Inference compute evaluation | Redress mechanism |
| | | Centralized documentation for downstream use |
| | | Documentation for responsible downstream use |

Figure 1: **Indicators.** The 100 indicators we use across 3 domains (upstream, model, and downstream) that are the same as in the October 2023 Foundation Model Transparency Index.
.

companies to submit *transparency reports*[5]: we reached out to leadership at 19 companies: 01.AI, Adept, AI21 Labs, Aleph Alpha, Amazon, Anthropic, BigCode/Hugging Face/ ServiceNow, Cohere, Databricks, Google, IBM, Inflection, Meta, Microsoft, Mistral, OpenAI, Stability AI, Writer, and xAI.[6] 14 developers agreed to prepare reports and designated their flagship foundation model.[7]

| Name | Flagship Model | Release | Input | Output | Status | Headquarters | WH1 | WH2 | FMF | AIA |
|---|---|---|---|---|---|---|---|---|---|---|
| Adept | Fuyu-8B | Open weights | T, I | T | Startup | USA | | | | |
| AI21 Labs [†] | Jurassic-2 | API | T | T | Startup | Israel | | | | |
| Aleph Alpha | Luminous Supreme | API | T, I | T | Startup | Germany | | | | |
| Amazon [†] | Titan Text Express | API | T | T | Big Tech | USA | ✓ | | | |
| Anthropic [†] | Claude 3 | API | T, I | T | Startup | USA | ✓ | | ✓ | |
| BC/HF[†]/SN | StarCoder | Open weights | T | T | Startup | USA | | | | ✓ |
| Google [†] | Gemini 1.0 Ultra API | API | T, I, A, V | T, I | Big Tech | USA | ✓ | | ✓ | |
| IBM | Granite | API | T | T | Big Tech | USA | | ✓ | | ✓ |
| Meta [†] | Llama 2 70B | Open weights | T | T | Big Tech | USA | ✓ | | | ✓ |
| Microsoft | Phi-2 | Open weights | T | T | Big Tech | USA | ✓ | | ✓ | |
| Mistral | Mistral 7B | Open weights | T | T | Startup | France | | | | |
| OpenAI [†] | GPT-4 | API | T, I | T | Startup | USA | ✓ | | ✓ | |
| Stability AI[†] | Stable Video Diffusion | Open weights | T | V | Startup | UK | | ✓ | | ✓ |
| Writer | Palmyra-X | API | T | T | Startup | USA | | | | |

Table 1: **Selected Foundation Model Developers.** Information on the 14 selected foundation model developers: the developer name, its flagship model, the release strategy for the model, the model's input and output modalities, the developer's corporate status, and the developer's headquarters. BC/HF/SN abbreviates BigCode/Hugging Face/ServiceNow. T, I, A, and V abbreviate text, image, audio, and video as modalities, respectively. [†] indicates the developer was evaluated in FMTI v1.0. We also indicate if developers were involved in the White House's voluntary commitments for the management of risks posed by AI announced in July 2023 (WH1), and commitments by additional organizations in the same areas announced in September 2023 (WH2) as well as if they are founding member of the Frontier Model Forum (FMF) or AI Alliance (AIA). Concurrent with the release of FMTI v1.1, as of May 20, 2024, Amazon and Meta have joined the FMF.

Table 1 describes the developers and their flagship foundation models. 8 of the 14 developers are hold-overs from FMTI v1.0.[8] Three developers are assessed for the same models as v1.0 (Jurassic-2 for AI21 Labs, Llama 2 for Meta, GPT-4 for OpenAI), whereas five are assessed for new models (Titan Text Express for Amazon, Claude 3 for Anthropic, StarCoder for BigCode/HuggingFace/ServiceNow, Gemini 1.0 Ultra API for Google, and Stable Video Diffusion for Stability AI). The six new developers and their models are Fuyu-8B for Adept, Luminous Supreme for Aleph Alpha, Granite for IBM, Phi-2 for Microsoft, Mistral 7B for Mistral, and Palmyra-X for Writer. In aggregate, the FMTI v1.1 composition has broader geographic coverage (e.g. from 0 to 2 companies headquartered in the European Union), broader modality coverage (e.g. Gemini takes audio and video as input), and a more even balance of open and closed foundation models (i.e. from 3 of 10 open models in v1.0 to 6 of 14 open models in v1.1).

### 3.3 Information gathering

In FMTI v1.0, Bommasani et al. (2023a) identified publicly-available sources of information for each developer through a systematic protocol for searching the Internet, which provided the information for all scoring decisions. This approach has four potentially undesirable properties. First, given that information is

---

[5]As we describe later, companies submitted an initial report that was modified through the FMTI v1.1 process. The final report that we publish is validated by the company but, therefore, different from this initial report. For brevity, we refer to both as transparency reports.

[6]FMTI v1.0 evaluated BigScience/Hugging Face, which together developed the BLOOMZ model; in FMTI v1.1 we evaluate BigCode/Hugging Face/ServiceNow, which together developed the StarCoder model. Throughout this paper we refer to BigCode/Hugging Face/ServiceNow as a single entity (i.e. the developer of StarCoder), though Hugging Face and ServiceNow are companies while BigCode is "an open scientific collaboration working on the responsible development and use of large language models for code" supported by ServiceNow and Hugging Face. See `https://www.bigcode-project.org/docs/about/mission/`.

[7]We provided guidance that the flagship foundation model should be "based on a combination of the following factors: greatest resource expenditure, most advanced capabilities, and greatest societal impact."

[8]Cohere and Inflection declined to participate in FMTI v1.1.

decentralized across the Internet, the researchers may have missed information.[9] Second, the relationship between a piece of public information and an indicator may be indirect and oblique, leading to greater subjectivity in scoring. Third, focusing on public information aligns with a developer's current level of transparency but does not provide developers with an opportunity to disclose further information. Finally, and most fundamentally, this search significantly adds to the cost of executing the index.

In FMTI v1.1, we request transparency reports from each developer that directly address each of the 100 indicators. This change in the information gathering process alters the dynamics for the four aforementioned considerations. First, if we assume developers are strongly incentivized to be their own best advocates and are certainly the most knowledgeable entities about their models, then the information they compile should be complete. Second, by having developers directly clarify information on indicators affirmatively, uncertainties that contributed to more subjective scoring are addressed. Third, by allowing developers to include information that was not-previously public, which is made public through this process, opportunities arise for greater transparency. Finally, by having developers gather information, the cost we bear is reduced.

### 3.4 Scoring

In FMTI v1.0, once information was identified, two researchers independently scored each of the 1000 (indicator, developer) pairs. The agreement rate was 85.2% (148 disagreements): in the event of disagreement, the researchers discussed and came to agreement. These initial scores were sent to developers to permit rebuttal: following a two-week rebuttal process, final scores were published in October 2023.

For FMTI v1.1, using the information identified through the developer-submitted transparency reports, two researchers independently scored each of the 1400 (indicator, developer) pairs. The standard of each indicator is the same as in FMTI v1.0: see Bommasani et al. (2023a, Appendix B) for the per-indicator scoring standard. The agreement rate was 85.3% (206 disagreements): in the event of disagreement, the researchers discussed and came to agreement.[10]

These initial scores were sent to developers to permit rebuttal: in contrast to FMTI v1.0, the rebuttal process was a more iterative multi-week process that involved email exchanges and video meetings. Through this correspondence, companies clarified existing information and disclosed new information, which is reflected in the final form of the transparency reports we publish. Ultimately, companies validated their transparency reports and approved their release. In doing so, unlike FMTI v1.0, these reports make explicit instances where (i) developers disclose useful information yet (ii) we felt the disclosure was insufficient to award a point.

### 3.5 Timeline

We summarize the execution of FMTI v1.1 as follows:

1. *Developer solicitation (December 2023 – January 2024).* We contacted leadership at 19 companies developing foundation models, requesting they submit transparency reports.

2. *Developer reporting (February 2024).* 14 developers designated their flagship foundation model and submitted transparency reports in relation to each of the 100 transparency indicators for their flagship model.

3. *Initial scoring (March 2024).* We reviewed the developers' reports, ensuring a consistent horizontal standard across all developers in terms of how each indicator was scored.

4. *Developer response (April 2024).* We returned the scored reports to developers, who then contested scores on specific indicators (potentially including the disclosure of additional information to justify a different score). Following this process, we finalized the transparency reports, which were validated by the developers prior to public release.

---

[9]However, this does beg the question of whether the information being public is truly constitutive of transparency if it is not discovered through a systematic and high-effort search.

[10]Future versions of FMTI may explore having independent scorers separate from the FMTI research team to increase reproducibility.

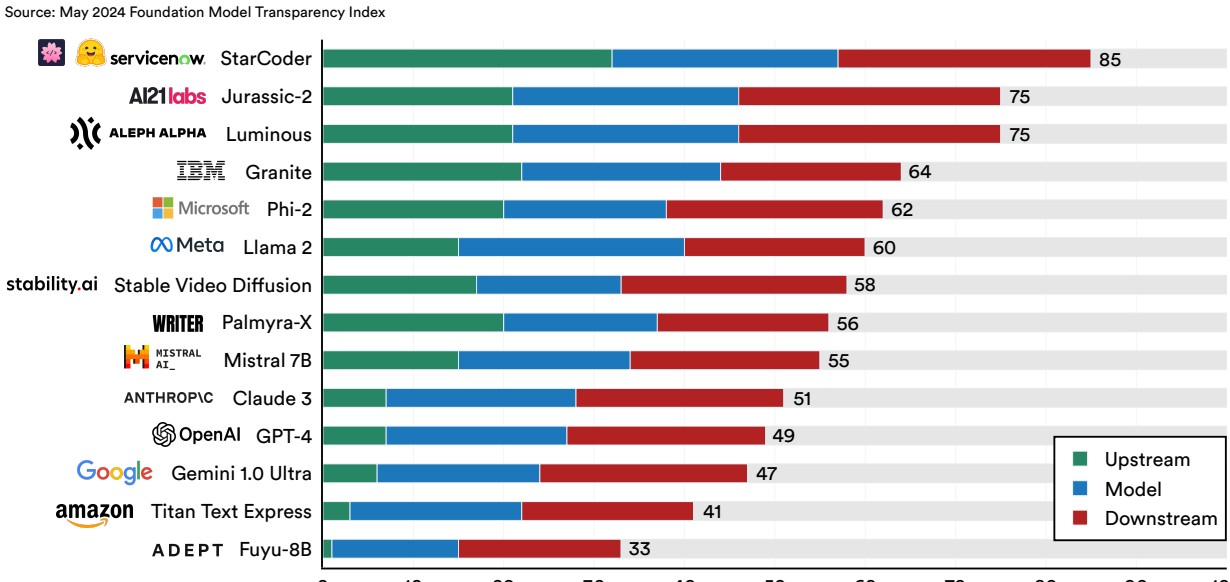

Figure 2: **Scores by Domain.** The overall scores disaggregated into the three domains: upstream, model, and downstream.

## 4 Results

We analyze this iteration of the Index on its own (§4.1), in relation to the first iteration (§4.2), and specifically in terms of new disclosures (§4.3).

### 4.1 Standalone results of FMTI v1.1

**While the average score on the Index has significant room for improvement, there is high variability among developers.** Based on the overall scores (right of Figure 1), 11 of the 14 developers score below 65, showing that there is a significant lack of transparency in the foundation model ecosystem and substantial room for improvement across developers. The mean and median are 57.93 and 57 respectively, with a standard deviation of 13.98. The highest-scoring developer scores points for 85 of the 100 indicators, while the lowest-scoring developer scores 33. The 3 top-scoring developers (BigCode/Hugging Face/ServiceNow, Aleph Alpha, and AI21 Labs) are more than one standard deviation above the mean, the next 9 are near the mean (IBM, Microsoft, Meta, Stability AI, Writer, Anthropic, OpenAI, Mistral, Google), and the 2 lowest-scoring developers are more than one standard deviation below the mean (Amazon, Adept).

**Improvement is feasible for each developer.** In spite of significant opacity, for 96 of the 100 indicators there exists some developer that scores points, and of these there are 89 where multiple developers score points. The disclosures that developers make to satisfy these indicators provide a concrete example of how all developers can be more transparent. If developers emulate the most-transparent developer for each indicator, overall transparency would improve sharply.

**Developers disclose significant new information, which contributes to their scores.** A developer's total score on the Foundation Model Transparency Index reflects the information that it discloses about its flagship foundation model in relation to each of the 100 indicators in the Index. In this version of the Index, we note where this information is disclosed as part of the process of conducting the Index (i.e. where developers disclosed the information for the first time via their report, or where developers updated their documentation in response to the Index process). This new information contributes to developer's overall scores: developers on average release new information related to 16.6 indicators, which improves the average

**Foundation Model Transparency Index Scores by Major Dimensions of Transparency, May 2024**
Source: May 2024 Foundation Model Transparency Index

| | ADEPT | AI21labs | ALEPH ALPHA | amazon | ANTHROP\C | ServiceNow / BigCode | Google | IBM | Meta | Microsoft | MISTRAL AI_ | OpenAI | stability.ai | WRITER | |
| | Fuyu-8B | Jurassic-2 | Luminous | Titan Text Express | Claude 3 | StarCoder | Gemini 1.0 Ultra | Granite | Llama 2 | Phi-2 | Mistral 7B | GPT-4 | Stable Video Diffusion | Palmyra-X | Average |
|---|---|---|---|---|---|---|---|---|---|---|---|---|---|---|---|
| Data | 0% | 60% | 40% | 0% | 10% | 100% | 0% | 60% | 40% | 40% | 20% | 20% | 40% | 50% | **34%** |
| Labor | 0% | 43% | 71% | 14% | 14% | 100% | 29% | 43% | 29% | 100% | 100% | 14% | 100% | 43% | **50%** |
| Compute | 14% | 86% | 100% | 0% | 14% | 100% | 14% | 100% | 71% | 57% | 14% | 14% | 43% | 86% | **51%** |
| Methods | 0% | 100% | 100% | 50% | 75% | 100% | 75% | 100% | 75% | 100% | 100% | 50% | 75% | 100% | **79%** |
| Model Basics | 83% | 100% | 100% | 83% | 50% | 100% | 83% | 100% | 100% | 100% | 100% | 50% | 100% | 100% | **89%** |
| Model Access | 100% | 67% | 100% | 67% | 67% | 100% | 67% | 67% | 100% | 100% | 100% | 67% | 100% | 33% | **81%** |
| Capabilities | 80% | 80% | 100% | 80% | 100% | 100% | 80% | 60% | 100% | 100% | 100% | 100% | 60% | 100% | **89%** |
| Risks | 0% | 57% | 57% | 43% | 86% | 100% | 43% | 71% | 71% | 29% | 14% | 57% | 14% | 14% | **47%** |
| Mitigations | 0% | 40% | 20% | 20% | 40% | 0% | 40% | 80% | 60% | 0% | 60% | 60% | 0% | 20% | **31%** |
| Distribution | 57% | 86% | 100% | 57% | 86% | 100% | 57% | 86% | 71% | 71% | 71% | 71% | 86% | 71% | **77%** |
| Usage Policy | 40% | 100% | 100% | 80% | 100% | 100% | 100% | 40% | 40% | 100% | 40% | 80% | 60% | 80% | **76%** |
| Feedback | 67% | 100% | 67% | 33% | 33% | 100% | 67% | 67% | 33% | 67% | 67% | 33% | 67% | 33% | **60%** |
| Impact | 29% | 29% | 29% | 0% | 14% | 14% | 29% | 0% | 14% | 0% | 14% | 14% | 14% | 14% | **15%** |
| **Average** | **36%** | **73%** | **76%** | **41%** | **53%** | **86%** | **53%** | **67%** | **62%** | **66%** | **62%** | **49%** | **58%** | **57%** | |

(Left axis label: Major Dimensions of Transparency)

Figure 3: **Scores by Major Dimensions of Transparency.** The fraction of achieved indicators in each of the 13 major dimension of transparency. Major dimension of transparency are large subdomains within the 23 subdomains.

score by 14.2 points. For example, AI21 Labs and Writer newly disclosed the carbon emissions associated with building their flagship foundation models (200-300 tCO2eq and 207 tCO2eq respectively). See §4.3 for further details.

**The upstream domain is the most opaque and the region where the most transparent developers distinguish themselves.** Breaking the results down by domain (Figure 2), developers performed best on indicators in the downstream domain, scoring 65% of available points overall in comparison to 61% on the model domain and 46% in the upstream domain. Developers scored worse across upstream indicators: of the 20 indicators where developers score highest, just one indicator (model objectives) is in the upstream domain. High-scoring developers often differentiate themselves in terms of their transparency in the upstream domain; the top scorer overall, BigCode/Hugging Face/ServiceNow, scored all 32 points, whereas the lowest scorer overall, Adept, scored just one point. The standard deviation of scores on the upstream domain (8.8) is more than double that of the other two domains (3.6 each), reflecting the much greater spread across the domain. On the whole, developers are less transparent about the data, labor, and compute used to build their models than how they evaluate or distribute their models. Prior work has emphasized the importance of these particularly opaque domains (Crawford, 2021; Gebru et al., 2018; Luccioni & Hernández-García, 2023).

**Scores varied across subdomains, with developers scoring best on user interface, capabilities, and model basics.** Disaggregating each domain, we consider the 23 subdomains with Figure 3 showing results for 13 major subdomains. Scores varied greatly across subdomains: 86 percentage points separate the average scores on the most transparent and least transparent subdomains. The subdomains with the highest scores are user interface (93%) capabilities (89%), model basics (89%), documentation for downstream deployers (89%), and user data protection (88%). Each of these subdomains is in the downstream domain, where developers scored near or above 70% on 6 of the 9 subdomains. These high scores were achieved in part through the release of new information. For instance, AI21 Labs released its first model card for Jurassic-2 during the FMTI v1.1 process, and Aleph Alpha and Stability AI made significant changes to their existing model cards. High scores on these subdomains reflects that disclosure in these areas is relatively less onerous for developers—disclosing documentation for downstream deployers as well as information about capabilities and model basics makes it easier and more appealing for customers to make use of a companies' flagship foundation model, meaning it is in developers' interest to do so.

**Data access, impact, and trustworthiness are the least transparent subdomains.** The subdomains with the lowest total scores are data access (7%), impact (15%), trustworthiness (29%), and model mitigations

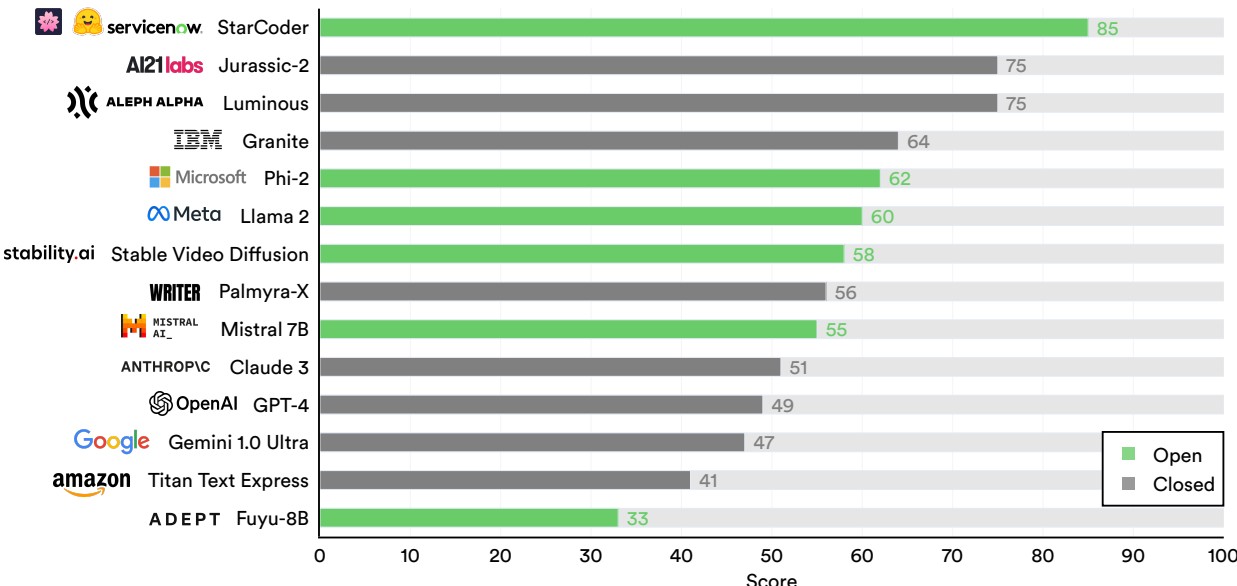

Figure 4: **Overall Scores by Release Strategy.** The overall scores for the 6 open developers (Adept, BigCode/Hugging Face/ServiceNow, Meta, Microsoft, Mistral, Stability AI) and the 8 closed developers (AI21 Labs, Aleph Alpha, Amazon, Anthropic, Google, IBM, OpenAI, Writer).

(31%). Developers score 50% or less on 10 of 23 subdomains in the index, including 3 of the 5 largest subdomains—impact (15%), data (34%), and data labor (50%). The lack of transparency in these subdomains shows that the foundation model ecosystem is still quite opaque—there is little information about how people use foundation models, what data is used to build foundation models, and whether foundation models are trustworthy.

**Open developers match closed developers on downstream indicators, and exceed them on upstream indicators.** Developers adopt different release strategies (Solaiman, 2023) for their flagship foundation models: six developers release open foundation models (Kapoor et al., 2024), meaning the model weights are widely available, whereas the other eight employ a more closed release strategy.[11] Open developers generally outperform closed developers (Figure 4: the median open developer scores 5.5 points higher than the median closed developer. While making the weights of a model openly available is correlated with greater overall transparency, it does not necessarily imply greater transparency about matters like data, compute, or usage.

The difference in transparency between open and closed developers is attributable to the substantial gap in upstream transparency: within the upstream domain, the median open developer scores 3 additional points on indicators in the upstream subdomain over the median closed developer. Within each subdomain, the median open developer scores as many or more points than the median closed developer on all but 5 of the 23 subdomains (i.e., risks, model mitigations, trustworthiness, distribution, usage policy, and model behavior policy). Open developers outscore closed developers by the widest margin on the following subdomains: data labor, data, and model access. Though open developers may struggle to gauge the downstream use of their models (Klyman, 2024), which closed developers may be able to directly monitor, the median open developer scores the same number of points on the downstream domain as the median closed developer.

---

[11]Aleph Alpha shares model weights for its flagship foundation model with some customers on premises, but this does not mean that the model weights are widely available and so it is considered a closed foundation model developer per the definition of Kapoor et al. (2024).

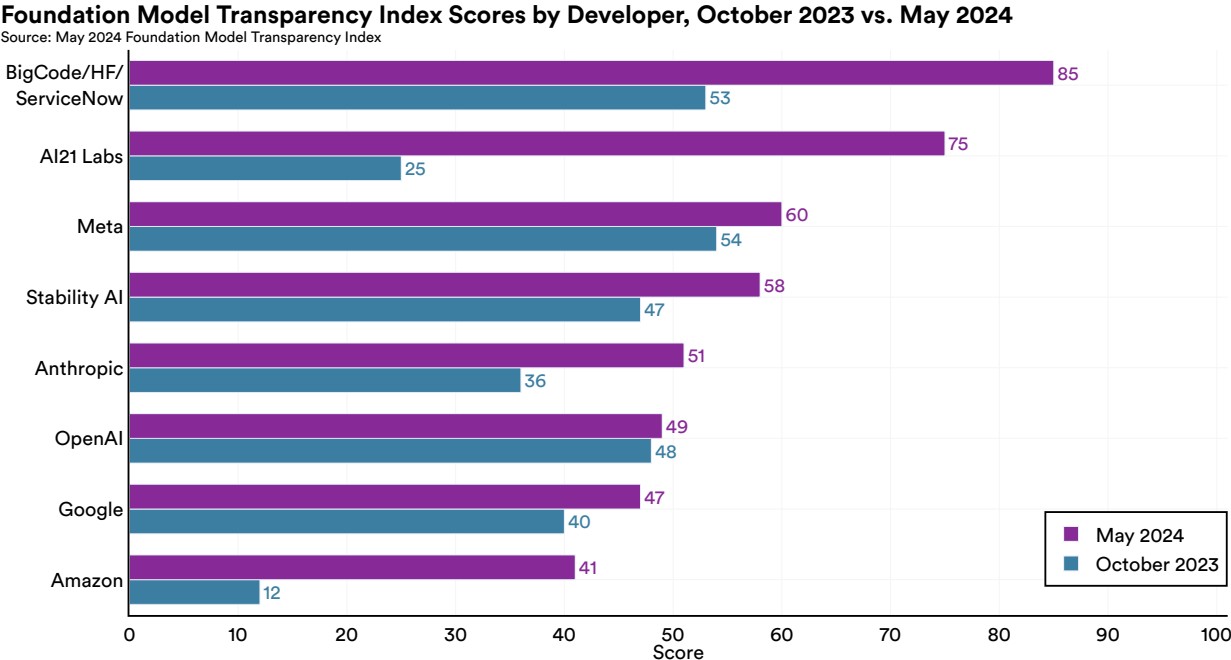

Figure 5: **Change in Overall Scores.** The FMTI v1.0 and v1.1 overall scores for the eight developers assessed in both versions.

Developers that are part of the AI Alliance (Hugging Face, IBM, Meta, ServiceNow, Stability AI), which often advocates for open releases of model weights, outscore developers that are founding members of the Frontier Model Forum (Anthropic, Google, Microsoft, OpenAI) by a median of 12 points, with higher average scores across upstream, model, and downstream indicators.[12]

Still, closed developers outperform open developers in several areas related to policies and enforcement. Closed developers generally share more information about if and how they enforce their policies regarding user and model behavior, outperforming open developers on these subdomains by 2 and 1 points respectively.[13] Closed developers also score higher on the risks and model mitigations subdomains, as several open developers do not describe or demonstrate risks associated with their flagship foundation model and closed developers are more likely to describe and demonstrate any risk mitigations that are taken at the model level.[14] The discrepancy in transparency between open and closed foundation model developers is a reflection of the current state of the ecosystem, not a fundamental reality about the transparency of developers that do or do not make the weights of their foundation models widely available.

### 4.2 Comparative results between FMTI v1.1 and v1.0

**Transparency increased across the board from v1.0 to v1.1.** Foundation model developers significantly improved their scores between October 2023 and May 2024, with the average score rising from 37 to 58 out of 100. Scores improved on every domain, with average upstream scores improving by the greatest margin (+7.6 points) followed by downstream (+7.2) and model (+6.1). As a result, there is significantly more

---

[12]The AI Alliance is a coalition of developers, universities, and researchers "who collaborate to advance safe, responsible AI rooted in open innovation." The Frontier Model Forum is an industry group whose founding members are Anthropic, Google, Microsoft, and OpenAI; the Frontier Model Forum committed to "advancing AI safety research," "identifying best practices," "collaborating across sectors," and "help[ing] AI meet society's greatest challenges." See `https://thealliance.ai/` and `https://www.frontiermodelforum.org/`.

[13]As with some other subdomains, the difference in scores here is driven by just one or two of the 14 developers.

[14]Open developers, however, outscore closed developers on data mitigation indicators. They often do not score points on model mitigation indicators because they do not explicitly state that no such mitigations were applied at the model level.

**Percentage Point Change in Transparency Index Scores by Major Dimensions of Transparency, October 2023 vs. May 2024**

Source: May 2024 Foundation Model Transparency Index

| | AI21 Labs | Amazon | Anthropic | BigCode/HF/ ServiceNow | Google | Meta | OpenAI | Stability AI |
|---|---|---|---|---|---|---|---|---|
| Data | +60% | +0% | +10% | +40% | -20% | +0% | +0% | +0% |
| Labor | +43% | +14% | -14% | +14% | +29% | +0% | +0% | +86% |
| Compute | +86% | +0% | +14% | +86% | +0% | +14% | +0% | -14% |
| Methods | +100% | +50% | +0% | +0% | +0% | +0% | +0% | -25% |
| Model Basics | +67% | +50% | -17% | +0% | +17% | +0% | +0% | +17% |
| Model Access | +33% | +33% | +33% | +0% | +33% | +0% | +0% | +0% |
| Capabilities | +20% | +60% | +20% | +20% | +0% | +40% | +0% | +20% |
| Risks | +29% | +43% | +57% | +100% | +14% | +14% | +0% | +0% |
| Mitigations | +40% | +0% | +0% | +0% | +0% | +0% | +0% | +0% |
| Distribution | +43% | +14% | +29% | +29% | -14% | +0% | +14% | +14% |
| Usage Policy | +80% | +60% | +40% | +80% | +40% | +0% | +0% | +20% |
| Feedback | +67% | +33% | +0% | +67% | +33% | +0% | +0% | +33% |
| Impact | +14% | +0% | +14% | +0% | +14% | +0% | +0% | +0% |

*Major Dimensions of Transparency*

Figure 6: **Change in Subdomain Scores From FMTI v1.0 to FMTI v1.1.** This figure shows the percentage point change in scores for major subdomains for the eight developers that are included in both the October 2023 and May 2024 versions of the Foundation Model Transparency Index.

information publicly available about the upstream resources developers use to build foundation models, the models themselves and how they are evaluated, and their downstream distribution and use.

**Transparency increased in nearly all subdomains from v1.0 to v1.1.** Developers improved their scores on every subdomain with the exception of data access. The largest improvements in subdomain scores were in compute (average increase of +2.4 indicators per developer), data labor (+2.3), and risks (+1.6). This broad improvement demonstrates that the overall trend in recent years toward reduced transparency is more nuanced than is commonly understood, though transparency is still lacking with respect to the data used to build foundation models and the impact they have once deployed.

**Transparency increased in subdomains that were especially opaque in v1.0.** Several of the areas of the index that were least transparent in v1.0 show significant improvement in v1.1, including subdomains such as compute, methods, risks, and usage policy. For example, the total score across companies for the compute subdomain rose from 17% in v1.0 to 51% in v1.1. Compute is potentially one of the most intractable areas for disclosure as it relates to the environmental impact of building foundation models—and therefore could be associated with additional legal liability for developers and deployers—yet we see improvement across compute indicators. This improvement is driven to a significant degree by new information that companies have disclosed, with companies disclosing new information on compute usage (6 companies disclosed new information), development duration (4), energy usage (4), compute hardware (3), hardware owner (3), and carbon emissions (2). In this way, transparency about compute expenditure has spillover effects for transparency about environmental impact, providing a potential model for translating information disclosure about one area into details about the impact of the foundation model supply chain.

Transparency has improved substantially with respect to companies' policies regarding acceptable use and behavior of their models. The total score across companies for the usage policy subdomain rose from to 44% to 57%, which was driven by increased transparency related to how these policies are enforced. Similarly, the total score across companies for the model behavior policy subdomain rose from 23% to 69%. While

increased transparency in these domains is a relatively light lift—as companies merely need to state whether they have such policies and, if so, describe how they are enforced—this form of transparency is not costless for companies as it highlights potential failure modes for their models (Klyman, 2024). Transparency about how companies restrict the use and behavior of their models can facilitate independent evaluation and red teaming due to reduced legal uncertainty, promoting research on safety and trustworthiness (Longpre et al., 2024a).

**Data remains a key area of opacity.** Several areas of the Index exhibit sustained and systemic opacity: almost all developers remain opaque on these matters. Developers display a fundamental lack of transparency with respect to data, building on frustrations of data documentation debt (Bandy & Vincent, 2021; Sambasivan et al., 2021). Transparency on the data subdomain rose from 20% to 34% from v1.0 to v1.1, but BigCode/Hugging Face/ServiceNow is the only developer that scores points on indicators relating to data creators, data copyright status, data license status, and personal information in data. Only 3 developers (Aleph Alpha, BigCode/Hugging Face/ServiceNow, IBM) score points on 6 or more of the 10 data indicators, while 6 developers score 2 or fewer points. These low scores reflect the ongoing crisis in data provenance (Longpre et al., 2024b; 2023a), wherein companies share no information about the license status of their datasets, preventing downstream developers from ensuring they are complying with such licenses.

While scores on data labor improved from 17% to 50% from v1.0 to v1.1, this was driven in large part by an increase in the number of companies that do not use data labor outside of their own organization (BigCode/Hugging Face/ServiceNow, Microsoft, Mistral). Considering only the 11 developers who do not disclose that they do not use external data labor, scores on the data labor subdomain fall to 36%, which would make it the sixth lowest scoring subdomain. Only one developer (Stability AI) that discloses its use of external data labor discloses the wages that it pays data laborers, highlighting how a lack of transparency may limit accountability for worker exploitation (Gray & Suri, 2019b).

Transparency in data access, one of the lowest scoring subdomains across developers in v1.0, declined in v1.1 from 20% to 7%. This reflects the significant legal risks that companies face associated with disclosure of the data they use to build foundation models. In particular, these companies may face liability if the data contains copyrighted, private, or illegal content such as child sexual abuse material (Lee et al., 2024; Solove, 2024; Thorn, 2024).

**Developers disclose little information about the real-world impact of their foundation models.** Developers still lack transparency about the real-world impact of their models, and any steps they take to mitigate negative impacts pre-deployment. Of the major dimensions of transparency in Figure 3, developers score worst on the impact subdomain (as they did in v1.0). Each of the four indicators where no developer scores points (affected market sectors, affected individuals, usage reports, and geographic statistics) are in the impact subdomain. This means that the public has little to no information about who uses foundation models, where foundation models are used, and for what purpose. The lack of transparency regarding these matters inhibits effective governance of foundation models, as it is difficult for governments or civil society organizations to pressure companies to responsibly deploy their models if there is no information about the impact of deployment. In many cases this opacity stems from a lack of information sharing between developers, deployers, and customers; developers generally do not know how their foundation model is being used unless a deployer monitors use or receives and shares information about use from its customer.

**The 8 developers evaluated in both October 2023 and May 2024 showed marked improvement, or 19 points on average.** For 3 of these developers we evaluate the same flagship foundation model (Jurassic-2 for AI21 Labs, Llama 2 for Meta, and GPT-4 for OpenAI) as FMTI v1.0 while for the other 5 we evaluate a different flagship foundation model (Titan Text Express for Amazon, Claude 3 for Anthropic, StarCoder for BigCode/Hugging Face/ServiceNow, Gemini 1.0 Ultra API for Google, and Stable Video Diffusion for Stability AI). All 8 companies' scores increased, as shown in (Figure 5): AI21 Labs (+50), BigCode/Hugging Face/ServiceNow (+32) and Amazon (+29) made the largest improvements.

**Developers that were assessed only in v1.1 performed slightly worse than those assessed in both v1.0 and v1.1.** The 6 developers that were assessed only in v1.1 (Adept, Aleph Alpha, IBM, Microsoft, Mistral, Writer) have slightly lower scores than those assessed in both v1.0 and v1.1. Their mean total score

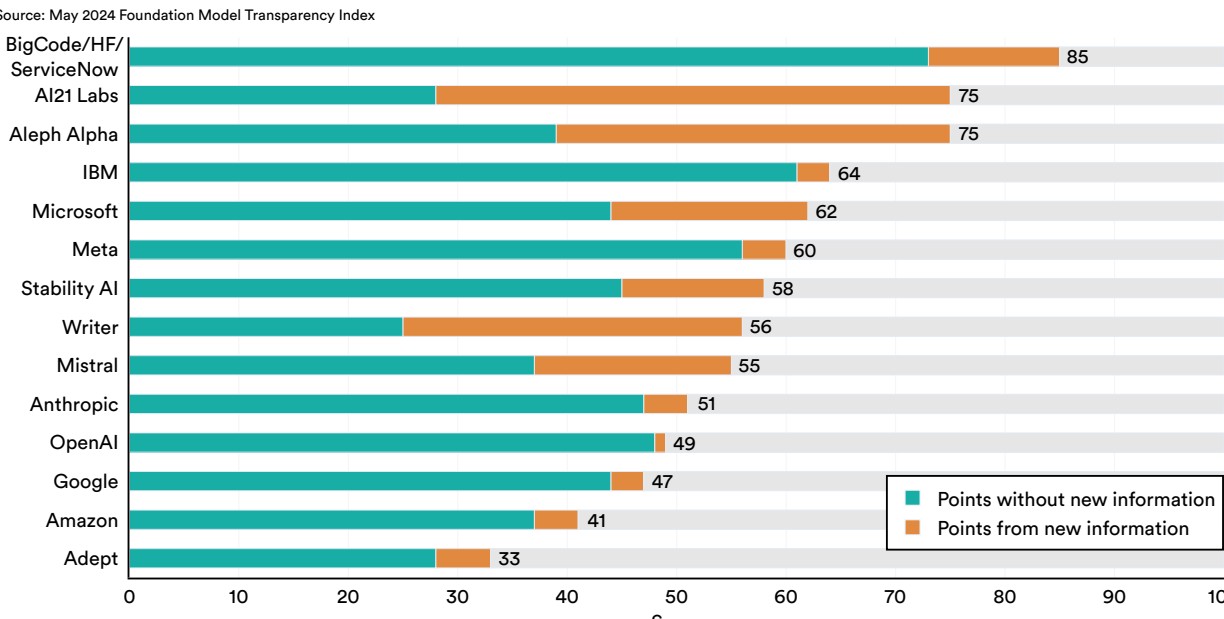

**Disclosure of New Information and Scores by Foundation Model Developer, May 2024**

Source: May 2024 Foundation Model Transparency Index

Figure 7: **Scores by New Information Status.** The overall scores disaggregated based on whether the information was newly disclosed.

is 57.5, which is 1 point lower than that of the other 8 developers.[15] Developers that were included in FMTI v1.0 had the benefit of having already evaluated their own transparency practices in relation to this initiative (i.e. in FMTI v1.0 they had the opportunity to rebut scores provided by Bommasani et al. (2023a), meaning it may have been relatively easier for them to compile transparency reports. The 6 developers assessed only in FMTI v1.1 include the lowest scoring developer and the two lowest scoring open model developers.

### 4.3 New information in FMTI v1.1

A key feature of the FMTI v1.1 methodology is that companies were able to disclose *new information*, meaning information that was not public at the onset of the FMTI v1.1 process. In some cases, this information is directly made public for the first time via the FMTI v1.1 transparency reports. In other cases, this information was incorporated into preexisting or new publicly available documentation from companies. For example, as we noted previously, AI21 Labs released the first model card for Jurassic-2 and Stability AI significantly updated the model card for Stable Video Diffusion.

**New information constitutes a large fraction of the score for several companies.** Figure 7 breaks down each developer's overall FMTI v1.1 scored based on which indicators were awarded for new information vs. information that was previously publicly available. For three developers (AI21 Labs, Aleph Alpha, Writer), new information constitutes roughly half the points awarded. In the case of Aleph Alpha, several new disclosures are made about data labor: all laborers are employed by Aleph Alpha, work in Germany, and are afforded labor protections as stipulated by German law. For Writer, new information is provided on compute: models are trained on 1024 NVIDIA A100 80GB GPUs for 74 days (910k GPU hours) on the Writer cluster, amounting to $8.2 \times 10^{23}$ FLOPs in compute, 812 MWh in energy, and 207tCO2eq in emissions.

**All developers disclose some new information.** In the case of OpenAI, the sole change to its disclosures from FMTI v1.0 is in relation to detecting machine-generated content. Specifically, OpenAI clarify that it "originally released a classifier that was taken down due to lack of accuracy. Our commitments are around audio / visual content for now, so this implies lack of ability to detect GPT-4 generated content".

---

[15]This is despite the fact that 4 of the 6 developers that are assessed only in FMTI v1.1 are open developers, which score higher on average.

**Foundation Model Developers' Disclosure of New Information by Major Dimension of Transparency, May 2024**
Source: May 2024 Foundation Model Transparency Index

Figure 8: **Aggregate New Information by Major Dimensions of Transparency.** The number of pieces of new information, aggregated across all developers, for each of the 13 major domains of transparency. Note: major dimensions of transparency each have a different numbers of indicators—the largest is data (10 indicators), followed by compute (7), distribution (7), impact (7), labor (7), risks (7), model basics (6), capabilities (5), mitigations (5), usage policy (5), feedback (3), methods (3), and model access (3).

In other cases, new information is disclosed to clarify existing information that was difficult to interpret based on publicly available documentation. For example, Amazon clarified how its model behavior policy and usage policy interoperate: "In the Bedrock user guide, we stated that AWS is committed to the responsible use of AI, and we use an automated abuse detection mechanisms to identify potential violations, we may request information about customers' use of Amazon Bedrock and compliance with our terms of service or a third-party provider's AUP. In the event that a customer is unwilling or unable to comply with these terms or policies, AWS may suspend access to Amazon Bedrock."

**New information drives much of the transparency for areas of large improvement.** Figure 8 totals the amount of new information across all developers for each major dimension of transparency. The most information is in the area of labor, which we noted previously is largely due to multiple companies clarifying they do not use data labor in building their flagship foundation models. Beyond this, other areas of large improvement from FMTI v1.0 to FMTI v1.1 are precisely those with large amounts of new information. Namely, more than 10% of the 233 pieces of new information are in the areas of compute and usage policy.

**Companies disclose new information for which they do not score points.** For example, several companies provide additional details about their data labor practices like general information about instructions to annotators and assurances that wages exceed local minimum wage. And on the indicator about data creators, which is awarded to only one of the 14 developers (BigCode/Hugging Face/ServiceNow), AI21 Labs discloses that "Most of the internet-connected population is from industrialized countries, wealthy, younger, and male, and is predominantly based in the United States." While these disclosures are insufficient for the stated standards for the associated indicators, we nonetheless emphasize that it may still be valuable for developers to make such information public. In a few cases, companies provide public justifications for why they do not disclose certain information. Most notably, Aleph Alpha states that "In line with the German copyright act ('Urheberrechtsgesetz'), data used for training is to be deleted after use and therefore can not be made available or distributed to external parties" in relation to the indicators about data access.

## 5 Discussion

Having characterized how transparency has changed from October 2023 to April 2024, we discuss how we reason about these changes (§5.1), what we recommend going forward (§5.2), and ways in which the Foundation Model Transparency Index could evolve in subsequent iterations (§5.3).

### 5.1 Interpretation of findings

There is significant room for improvement in the transparency of foundation model developers. Developers on average score just 58 out of 100, with major gaps in multiple subdomains for most developers. Developers' transparency reports are incomplete, lacking disclosures on many important matters.

Nevertheless, our findings provide significant reasons for optimism about the prospects for improved transparency. Foundation model developers shared a significant amount of new information about how they build, evaluate, and deploy their models via FMTI v1.1. Some of the areas of the index that were least transparent in v1.0 show significant improvement in v1.1, including subdomains such as compute, methods, risks, and usage policy. Several companies have become much more transparent, with some releasing model cards or other documentation for their flagship foundation models for the first time. By engaging directly with foundation model developers, we have shown that many firms are willing to disclose more information about some of their most powerful technologies.

Still, the overall state of transparency in the foundation model ecosystem remains poor. Developers are opaque about the data, labor, and compute used to build their models, they often do not release reproducible evaluations of risks or mitigations, and they do not share information about the impact their models are having on users or market sectors. Transparency in the foundation model ecosystem has been declining for several years, and this trend is unlikely to reverse in the near term.

Where developers do disclose information, they sometimes disclose information for only the least onerous indicators within a subdomain. For example, in the risks subdomain, most developers describe risks (12 of 14 developers) and many demonstrate risks (8 of 14). However, fewer developers evaluate unintentional harm (5 of 14) or intentional harm (4 of 14). We see the same trends for model mitigations and data labor, where developers disclose information regarding less intensive indicators but no others.

**Why do developers not disclose specific information?** Based on experiences engaging with these developers, several reasons were specifically put forth to justify why the developer would not disclose information in relation to a specific indicator. In some cases, developers cited countervailing interests that they found to be in tension with transparency (e.g. legal exposure from data transparency, trade secrets for model details). In these cases, we will reflect on how the FMTI can first yield Pareto improvements, meaning improvements in transparency at no cost to these other interests by having well-scoped definitions for indicators, before broaching areas of clearer trade-offs. Critically, we emphasize that the aforementioned cases generally indicate (i) the developer internally has the desired information but (ii) does not want to disclose this information externally to the public. In contrast, in other cases we found there were cases where the developer simply did not have the desired information and producing it would be costly. In these cases, we understood that internal cross-team coordination, especially for large companies, was a key barrier: consultation across lawyers, engineers, product managers, and executives before authorizing release imposed significant practical barriers to information disclosure. Or, in a few cases, developers disclosed the barrier was the lack of information from an external party (e.g. energy details from computer providers, usage information from downstream users). Overall, these explanations illustrate the concrete factors, to our knowledge, that limit overall transparency of foundation model developers.

### 5.2 Recommendations

We present recommendations aimed at different stakeholders based on the findings of FMTI v1.1 as well as changes in the world over the past six months, building on a more extensive set of recommendations made in Bommasani et al. (2023a, §8).

### 5.2.1 Foundation model developers

As part of FMTI v1.1, we publish transparency reports that consolidate information disclosures from developers and that we release subject to their validation. In light of voluntary codes of conduct promulgated by the White House and the G7 that include commitments for foundation model developers to release transparency reports, we envisage the reports we release as part of FMTI v1.1 as rudimentary forms of such transparency reports (Bommasani et al., 2024). In addition, many of the areas where transparency is lacking in foundation models mirror those in previous waves of digital technology as well as in other industries such as finance and healthcare. For example, shortcomings in downstream transparency by foundation model developers mirror issues faced by social media companies in the last decade (Aspen Institute, 2021). For such indicators, investing in the expertise of trust and safety professionals could help inform how developers develop internal and external policies, including those related to transparency. As one example, social media platforms like Facebook, YouTube, and TikTok detail usage policy violations and government requests for user data in their transparency reports (Narayanan & Kapoor, 2023). Similarly, social media platforms often have well-established processes for communicating with users about account restrictions and handling user appeals (Trust and Safety Professional Association, 2023). Foundation model developers can adopt these policies and processes to improve their downstream transparency (Klyman, 2024).

### 5.2.2 Customers of foundation models

Purchasers (e.g. downstream developers, enterprise users of chatbots, or government entities) can exert negotiating power in procurement to increase transparency. Notably, some foundation model developers stated that their participation was driven by requests from customers to understand the transparency of their products. Others mentioned that their practices related to transparency with their customers are much better compared to the data they can share publicly—an example of the influence customers can have on business practices. The Foundation Model Transparency Index provides a structured way for customers to advance transparency from foundation model developers—both in terms of the information developers share with their clients and with the broader public.

In addition, governments can play a dual role as customers of foundation models (Quay-de la Vallee et al., 2024). As influential (and lucrative) procurers of technology, this can help them play a standard-setting role. For example, the U.S. government is one of the largest purchasers of various goods and services, which allows it to set the standards for how these goods and services are sold, shaping business practices across industries (Vinsel, 2019), including around transparency. While requirements on transparency by governments may lead to legal concerns around government overreach, such as concerns in the US related to the first amendment implications of compelled speech (Bankston & Hodges, 2024), standard-setting via procurement circumvents these concerns by relying solely on the voluntary commitment to these standards by model developers who want to enter a contract.

### 5.2.3 Transparency advocates

The transparency reports we release can enable transparency advocates in academic and civil society organizations to better understand developer practices. While for the purposes of the Index scores we set a threshold for constitutes sufficient disclosure to award a point, the underlying disclosures are considerably richer. We encourage researchers and journalists to investigate this information, which includes considerable variation across companies that we do not explore in this paper.

### 5.2.4 Policymakers

Policymakers can use our results to identify areas of pervasive opacity—including areas with sustained opacity (across FMTI v1.0 and v1.1) as well as areas with systematic opacity (across the developers we score). This can also highlight perverse business incentives that might lead to such a lack of transparency, and in turn, can inform regulation that addresses them. For example, sharing information about the data used to train foundation models might open up companies to liability concerns, such as due to the legal uncertainty around copyright violations. Regulation intended to address such perverse incentives, such as mandatory disclosure of training data, could help address these impediments to transparency.

Various policy efforts in the last two years have focused on addressing transparency in the foundation model ecosystem, including Canada's Code of Conduct on the Responsible Development and Management of Advanced Generative AI Systems, the EU AI Act, and the US Executive Order on the Safe, Secure, and Trustworthy Development and Use of Artificial Intelligence. Still, Bommasani et al. (2024) find that these efforts can lack specificity. Our iterative process with model developers provides an insight into how such specificity might arise in practice—government bodies might consider developing the institutional capacity to coordinate discussions with companies to ensure that they adhere to the standard required by a certain policy.

### 5.3 Next steps

We plan to conduct future versions of the Foundation Model Transparency Index on a regular basis. As stated by Bommasani et al. (2023c), as part of future iterations we will make changes to the Foundation Model Transparency Index to reflect changes in the foundation model ecosystem and the organizations that build and deploy foundation models.

For example, moving forward, the Foundation Model Transparency Index may change the indicators or their thresholds. For the purposes of clear comparisons, the indicators and scoring thresholds are the same across FMTI v1.0 and v1.1. Due to ambiguity (as evidenced by developers misinterpreting indicators), one possible change could include splitting existing indicators into multiple different ones in order to more clearly delineate the information required in each indicator. Due to saturation (as evidenced by most developers satisfying many indicators), possible changes could include increasing the scoring thresholds for existing indicators as well as introducing new indicators about information that is not currently assessed by the Index. And due to changes external to FMTI in the foundation model ecosystem, such as growing interest in the encoding of human values (Scherrer et al., 2024) and the implementation of reward models for aligning foundation models (Lambert et al., 2024), possible changes could include adding indicators that specifically reflect transparency in the values encoded in models as well as information about how reward models are trained and operationalized. Most fundamentally, while the initial 100 indicators were decided upon by Bommasani et al. (2023a) with guidance from others in the community, a more open-ended process for community-driven indicator proposals may be implemented.

## 6 Limitations

In general, transparency as a construct and indices as an approach have well-known limitations (Birchall, 2014; Valdovinos, 2018; Alloa & Thomä, 2018; Alloa, 2018; Sagar & Najam, 1998; Boldin, 1999; Santeramo, 2017; Greco et al., 2019; Schlossarek et al., 2019). There are also a number of well-known limitations of transparency in AI specifically (Ananny & Crawford, 2018b; Phang et al., 2022; Hartzog, 2023). These limitations are discussed at length by Bommasani et al. (2023a) and they apply equally to FMTI v1.1.

In §2.2, Bommasani et al. (2023a) write: "Transparency is far from sufficient on its own and it may not always bring about the desired change (Corbett & Denton, 2023). Salient critiques of transparency include:

- Transparency does not equate to responsibility. Without broad based grassroots movements to exert public pressure or concerted government scrutiny, organizations often do not change bad practices (Boyd, 2016; Ananny & Crawford, 2018a).

- Transparency-washing provides the illusion of progress. Some organizations may misappropriate transparency as a means for subverting further scrutiny. For instance, major technology companies that vocally support transparency have been accused of *transparency-washing*, whereby "a focus on transparency acts as an obfuscation and redirection from more substantive and fundamental questions about the concentration of power, substantial policies and actions of technology behemoths" (Zalnieriute, 2021).

- Transparency can be gamified. Digital platforms have been accused of performative transparency, offering less insightful information in the place of useful and actionable visibility (Ghosh & Faxon,

2023; Mittelstadt, 2019). As with other metrics, improving transparency can be turned into a game, the object of which is not necessarily to share valuable information.[16]

- Transparency can inhibit privacy and promote surveillance. Transparency is not an apolitical concept and is often instrumentalized to increase surveillance and diminish privacy (Han, 2015; Mohamed et al., 2020; Birchall, 2021). For foundation models, this critique underscores a potential tension between adequate transparency with respect to the data used to build foundation models and robust data privacy.

- Transparency may compromise competitive advantage or intellectual property rights. Protections of competitive advantage plays a central role in providing companies to the incentives to innovate, thereby yielding competition in the marketplace that benefits consumers. Consequently, work in economics and management studies have studied the interplay and potential trade-off between competitive advantage and transparency (Bloomfield & O'Hara, 1999; Granados & Gupta, 2013; Liu et al., 2023), especially in the discourse on corporate social responsibility [(Wu et al., 2020; Yu et al., 2018)].

Transparency is not a panacea. In isolation, more information about foundation models will not necessarily produce a more just or equitable digital world. But if transparency is implemented through engagement with third-party experts, independent auditors, and communities who are directly affected by digital technologies, it can help ensure that foundation models benefit society."

In §9.2, Bommasani et al. (2023a) address how several of these limitations of transparency and indices apply in the context of the Foundation Model Transparency Index. These limitations include equating transparency with responsibility, transparency washing, gaming the index, binary scoring, focusing on language models, and focusing on companies headquartered in the US. Beyond these limitations, we specifically discuss additional considerations that arose in this version.

**Considering only publicly available information.** At present, the Foundation Model Transparency Index exclusively considers public disclosure of information: such disclosures provide transparency to all stakeholders and are verifiable by the researchers conducting the Index. However, intermediary forms of information disclosure exist between no disclosure to entities beyond the developer and disclosure to everyone. The EU AI Act identifies core types of intermediary information disclosure: disclosure to the government (see Annex IX A) and disclosure to clients downstream in the foundation model supply chain (see Annex IX B). Kolt et al. (2024) provide initial guidance on stratified disclosures approaches for cybersecurity and biosecurity. Assessing companies' release of information via such intermediary forms of disclosure could encourage such disclosures, which can be beneficial even in the absence of public disclosures.

**Requiring that disclosure be explicit.** In order for a developer to score points on a given indicator, it must make an explicit disclosure related to that indicator. For example, scoring points on the hardware owner indicator requires that a developer explicitly state which organization(s) owns the primary hardware used in building the model; even if the developer considers this to be obvious and not worth stating in its documentation, it must disclose the owner explicitly in its transparency report to score the point. We use this standard to promote reproducibility and remove ambiguity: rather than making assumptions based on what developers imply about their flagship foundation models, explicit disclosures ensure that anyone could re-score the developer in the same way.

**Company selection of flagship foundation models.** As part of a change to our methodology, for FMTI v1.1 each company selected its flagship foundation model to be assessed. While we provided guidance to companies regarding how to do so—we stated it should be based on a combination of resources expended, capabilities, and societal impact—we do not have sufficient insight into a company's operations (or, for example, what model sits behind an API) to validate that the model a developer designates as its flagship model is in fact its most significant model at present. This flexibility introduces risks, as it is possible that a developer might chose the most transparent of its foundation models in order to increase its score. This limitation stems in large part from the fact that we assess a developer's transparency based on a single flagship foundation model.

---

[16]According to Goodhart's Law, "when a measure becomes a target, it ceases to be a good measure" (Goodhart, 1984).

**Companies submitted transparency reports.** In this iteration of the Index, companies submit transparency reports to disclose key information about their foundation models. This comes with several limitations. Where a developer does not disclose information related to a particular indicator or does not affirmatively and explicitly disclose that it satisfies the indicator, we take this at face value and score that indicator as a zero. In FMTI v1.0, however, there were multiple cases that identified information in companies' documentation that they themselves did not believe satisfied a particular indicator. The methodology used in FMTI v1.1 prevents this from happening, instead assuming that companies know their own documentation best.

**No distinction between B2B and B2C companies.** The developers that we assess include companies with a variety of different business models, including those that primary develop models for enterprise customers (e.g. Aleph Alpha and Amazon) and those with a large number of individuals as customers (e.g. Anthropic and OpenAI). These differences in business models result in different approaches to transparency. B2B companies prioritize disclosing information directly to enterprise customers, who may be less concerned about public facing transparency. We use the same set of 100 transparency indicators for each company, which limits our ability to capture the nuances in how companies conceive of transparency and its value for their customers. A number of companies requested that we assess certain indicators as "Not Applicable" in light of their business model, but our methodology of binary scoring prevented us from contemplating this option.

**Low bar for awarding points.** Bommasani et al. (2023a) indicate "We were generally quite generous in the scoring process. When we determined that a developer scored some version of a half-point, we usually rounded up. Since we assess transparency, we award developers points if they explicitly disclose that they do not share information about a particular indicator. We also read developers' documents with deference where possible, meaning that we often awarded points where there are grey areas. This means that developers' scores may actually be higher than their documentation warrants in certain cases as we had a low bar for awarding points on many indicators." In future iterations of the index we may raise the threshold for scoring points (e.g. round down scores on indicators that might otherwise be half-points) as doing so would require a greater degree of transparency and potentially foster a more consistent grading standard.

## 7  Conclusion

The societal impact of foundation models is escalating, attracting the attention of firms, media, academia, government, and the public. The Foundation Model Transparency Index continues to find that transparency ought to be improved in this nascent ecosystem, with some positive developments since October 2023. By dissecting what developers do and do not publicly disclose, and how this has changed, the Index allows different stakeholders (e.g. developers, customers, investors, policymakers) to make more clear-eyed decisions. And, in turn, by establishing the practice of transparency reporting for foundation models, the Index surfaces a new resource that downstream developers, researchers, and journalists should capitalize on to build collective understanding. Moving forward, we hope that headway on transparency will demonstrably translate to better societal outcomes like greater accountability, improved science, increased innovation, and better policy.

**Acknowledgements.**  We thank the FMTI Advisory Board (Arvind Narayanan, Daniel E. Ho, Danielle Allen, Daron Acemoglu, Rumman Chowdhury) for their feedback and guidance. We thank Zak Rogoff for discussions and feedback related to this work. We especially thank Loredana Fattorini for her extensive work on the visuals for this project, as well as Shana Lynch for her work in publicizing this effort.

**Foundation Model Developers.**  We thank the following individuals at their respective organizations for their engagement with our effort: We emphasize that **this acknowledgement should not be understood as an endorsement of any kind by these individuals**, but simply that they were involved in our engagement with their organizations.

- Adept — David Luan, Kelsey Szot, Maxwell Nye, Augustus Odena

- Aleph Alpha — Jonas Andrulis, Joshua Kraft, Yasser Jadidi, Samuel Weinbach

- AI21 Labs — Yoav Shoham, Ori Goshen, Shanen J. Boettcher

- Amazon — Peter Hallinan, Sara Liu

- Anthropic — Jack Clark, Ashley Zlatinov, Frances Pye

- BigCode/Hugging Face/ServiceNow — Clement Delangue, Yacine Jernite, Sean Hughes, Leandro Von Werra, Harm de Vries, Irene Solaiman, Carlos Muñoz Ferrandis

- Google — James Manyika, Alexandra Belias, Kathy Meier-Hellstern, Reena Jana, Christina Butterfield, Will Hawkins, Dawn Bloxwich, Eve Novakovic, Victoria Langston, Demetra Brady

- IBM — Kush Varshney, Maryam Ashoori, Steven Sawyer

- Meta — Joelle Pineau, Melanie Kambadur, Esteban Arcaute, Joe Spisak, Manohar Paluri, Ragavan Srinivasan

- Microsoft — Eric Horvitz, Sebastien Bubeck, Suriya Gunasekar

- Mistral — Arthur Mensch, William El Sayed, Nicolas Schuhl

- OpenAI — Miles Brundage, Lama Ahmad

- StabilityAI — Ben Brooks, Carlos Riquelme, Paulo Rocha

- Writer — May Habib, Apara Sivaraman, Waseem AlShikh, Rowan Reynolds

**Conflict of Interest.** Given the nature of this work (e.g. potential to significantly impact particular companies and shape public opinion), we proactively bring attention to any potential conflicts of interest, deliberately taking a more expansive view of conflict of interest to be especially forthcoming.

- Betty Xiong is not, and has not, been affiliated with any of the companies evaluated in this effort or any other private sector entities.

- Kevin Klyman is not, and has not, been affiliated with any of the companies evaluated in this effort.

- Nestor Maslej is not, and has not, been affiliated with any of the companies evaluated in this effort or any other private sector entities.

- Percy Liang was a post-doc at Google (September 2011–August 2012), a consultant at Microsoft (May 2018–May 2023), and a co-founder of Together AI (July 2022–present). He is not involved involved in any of the companies evaluated in this effort.

- Rishi Bommasani is not, and has not, been affiliated with any of the companies evaluated in this effort.

- Sayash Kapoor worked at Meta until December 2020. He has not since worked for the company.

- Shayne Longpre worked at Google in 2022, and on BigCode-affiliated projects in 2023.

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

# A    Selection decisions

In conducting the index, core structural design decisions are (i) the indicators used to assess developers and (ii) the developers that are assessed. Here, we clarify both matters.

## A.1    Indicator selection

We use the same 100 indicators as FMTI v1.0 to facilitate direct comparison. These indicators are listed by domain in Figure 1.

## A.2    Developer selection

We contacted 19 foundation model developers to request these developers submit transparency reports for the purpose of conducting FMTI v1.1. Consistent with the principles used by Bommasani et al. (2023a), we only considered developers that are companies and that develop prominent foundation models. Specifically, we contacted leadership via email at 01.ai, Adept, AI21 Labs, Aleph Alpha, Amazon, Anthropic, BigCode (Hugging Face and ServiceNow), Cohere, Databricks, Google, IBM, Inflection, Meta, Microsoft, Mistral, OpenAI, Stability AI, Writer, and xAI. Following this email correspondence, and further clarification of the nature of the request, 14 foundation model developers agreed to provide the requested transparency reports.

Therefore, our selection process deliberately excluded foundation model developers that are not companies, even if they develop prominent foundation models, such as the Allen Institute for AI (developer of models such as OLMo; Groeneveld et al., 2024) and EleutherAI (developer of Pythia; Biderman et al., 2023). While developers such as AI2 and EleutherAI are often leaders in various types of transparency, releasing detailed information about data (Gao et al., 2020; Soldaini et al., 2024), evaluations (Gao et al., 2021; Magnusson et al., 2023), and the model development pipeline (Black et al., 2022; Biderman et al., 2023; Groeneveld et al., 2024), we consider only companies in selecting developers to assess in FMTI v1.1.

Our selection process also did not involve engagement with developers where we lacked connections with their leadership, which often coincides with models developed outside the United States and the Western hemisphere (e.g. the developers of Falcon (Almazrouei et al., 2023), Qwen (Bai et al., 2023), DeepSeek (DeepSeek-AI et al., 2024), and HyperCLOVA (Yoo et al., 2024)).

Finally, we did not engage any developer that released its first prominent foundation model during our execution of FMTI v1.1 (such releases include Apple's MM1 (McKinzie et al., 2024), SambaNova's Samba 1 (SambaNova, 2024), Reka's Reka Core (Team et al., 2024), and Snowflake's Arctic-1 (Merrick et al., 2024)). Moving forward, having successfully demonstrated that prominent foundation model developers have cooperated by submitting transparency reports, subsequent versions of the Foundation Model Transparency Index may engage these companies as well as others.

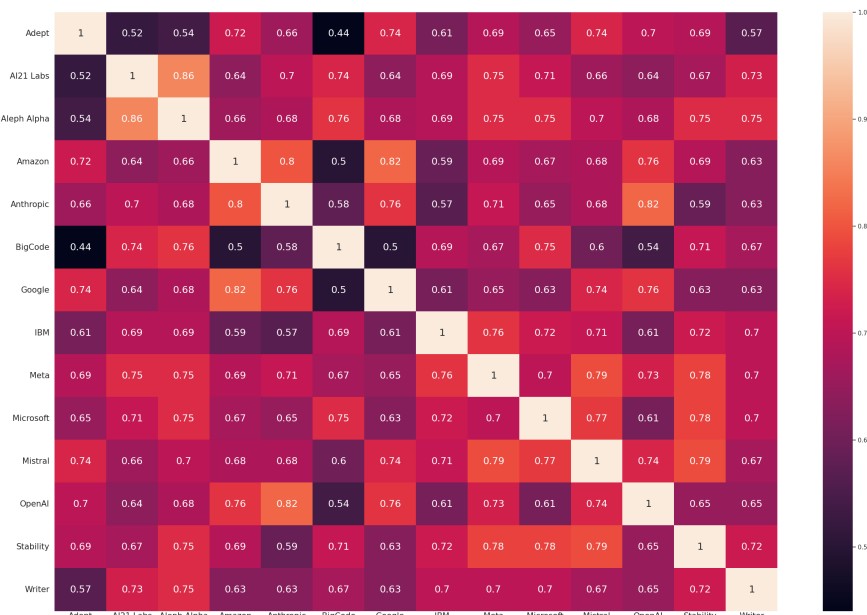

Figure 9: **Correlations between Companies.** The correlation between the scores for pairs of companies across all indicators. Correlation is measured using the simple matching coefficient (i.e. agreement rate), which is the fraction of all indicators for which both companies receive the same score (i.e. both receive the point or both do not receive the point).

## B    Extended Results

We present additional results, drawing inspiration from the analyses conducted in the first version of the Foundation Model Transparency Index.

### B.1    Developer correlations

**Measuring correlations.**    The $100 \times 14$ matrix of scores introduces data-driven structure. In particular, it clarifies relationships that arise in practice between different regions of the index. Here, we consider the *correlations*, in scores, focusing on company-to-company similarity for simplicity. For example, this analysis helps address the following: if two companies receive similar aggregate scores, is this because they satisfy all the same indicators or do they score points on two very different sets of indicators?

In Figure 9, we plot the correlation between every pair of companies. To measure correlation, we report the simple matching coefficient (SMC) or the agreement rate. The SMC is the fraction of the 100 indicators for which both companies receive the same score (i.e. both receive a zero or both receive a 1). As a result, a SMC of 0 indicates there is no indicator such that both companies receive the same score and a SMC of 1 indicates that for all indicators both companies receive the same score. For this reason, the correlation matrix is symmetric and guaranteed to be 1 on the diagonal.

**Upstream correlations.**    In Figure 10, we plot the correlation between every pair of companies when considering only indicators from the upstream domain.

**Model correlations.**    In Figure 11, we plot the correlation between every pair of companies when considering only indicators from the model domain.

**Downstream correlations.**    In Figure 12, we plot the correlation between every pair of companies when considering only indicators from the downstream domain.

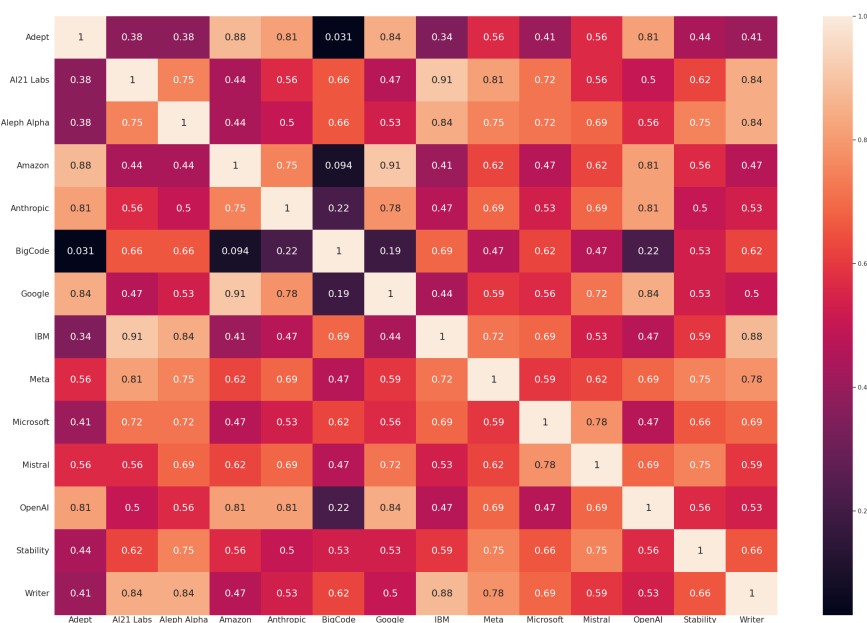

Figure 10: **Correlations between Companies (Upstream Indicators).** The correlation between the scores for pairs of companies across all indicators when only considering upstream indicators. Correlation is measured using the simple matching coefficient (i.e. agreement rate), which is the fraction of all indicators for which both companies receive the same score (i.e. both receive the point or both do not receive the point).

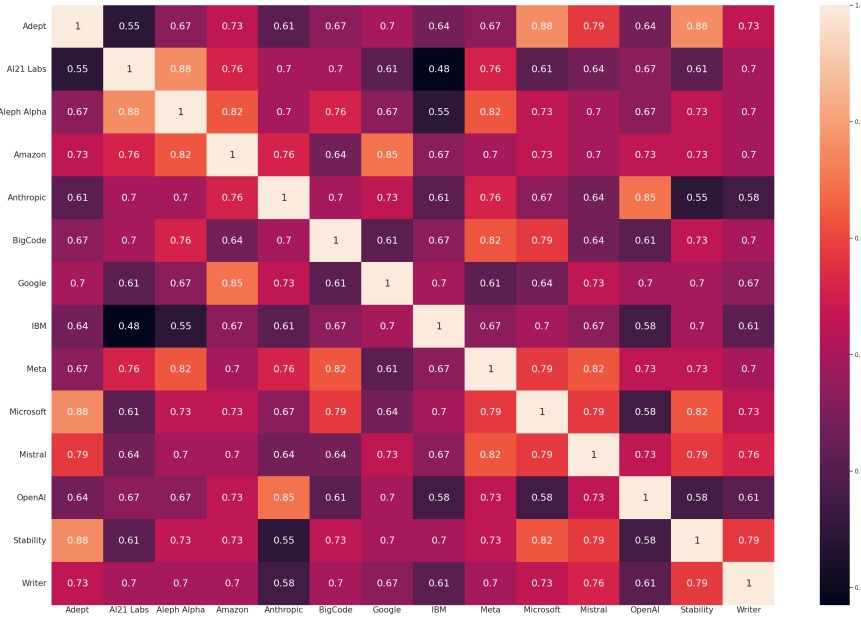

Figure 11: **Correlations between Companies (Model Indicators).** The correlation between the scores for pairs of companies across all indicators when only considering model indicators. Correlation is measured using the simple matching coefficient (i.e. agreement rate), which is the fraction of all indicators for which both companies receive the same score (i.e. both receive the point or both do not receive the point).

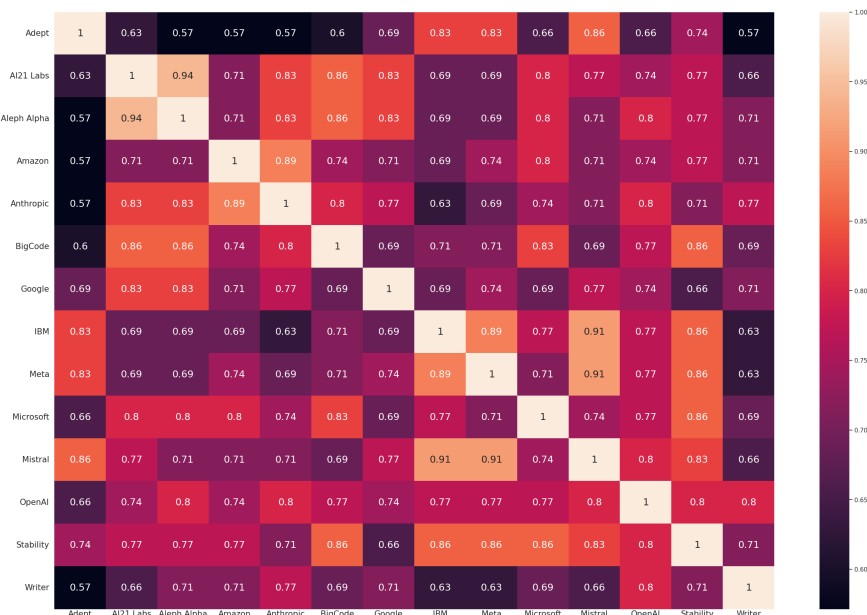

Figure 12: **Correlations between Companies (Downstream Indicators).** The correlation between the scores for pairs of companies across all indicators when considering only downstream indicators. Correlation is measured using the simple matching coefficient (i.e. agreement rate), which is the fraction of all indicators for which both companies receive the same score (i.e. both receive the point or both do not receive the point).

## B.2 Indicator-level results

**Assessing companies at the indicator level.** The core of the Foundation Model Transparency Index is the 100 indicators of transparency, which we aggregate into subdomains and domains in order to facilitate discussion and analysis of our results. We score each developer on each indicator (either 0 or 1) based on the information disclosed by the developer in relation to that indicator; each indicator is accompanied by a definition, which describes the indicator, and notes, which provide details about how that indicator is scored (Bommasani et al., 2023a, Appendix B). Below we provide each developers' score on every indicator, broken down by domain (i.e. upstream, model, and downstream).[17]

**Upstream Indicators.** In Figure 13, we show the scores of every developer on each of the indicators in the upstream domain. We also disaggregate uptream indicators by subdomain (data, data labor, data access, compute, methods, and data mitigations).

**Model Indicators.** In Figure 14, we show the scores of every developer on each of the indicators in the model domain. We also disaggregate model indicators by subdomain (distribution, usage policy, model behavior policy, user interface, user data protection, model updates, feedback, impact, and downstream documentation).

**Downstream Indicators.** In Figure 15, we show the scores of every developer on each of the indicators in the downstream domain. We also disaggregate downstream indicators by subdomain (model basics, model access, capabilities, limitations, risks, model mitigations, trustworthiness, and inference).

---

[17]https://www.github.com/stanford-crfm/fmti

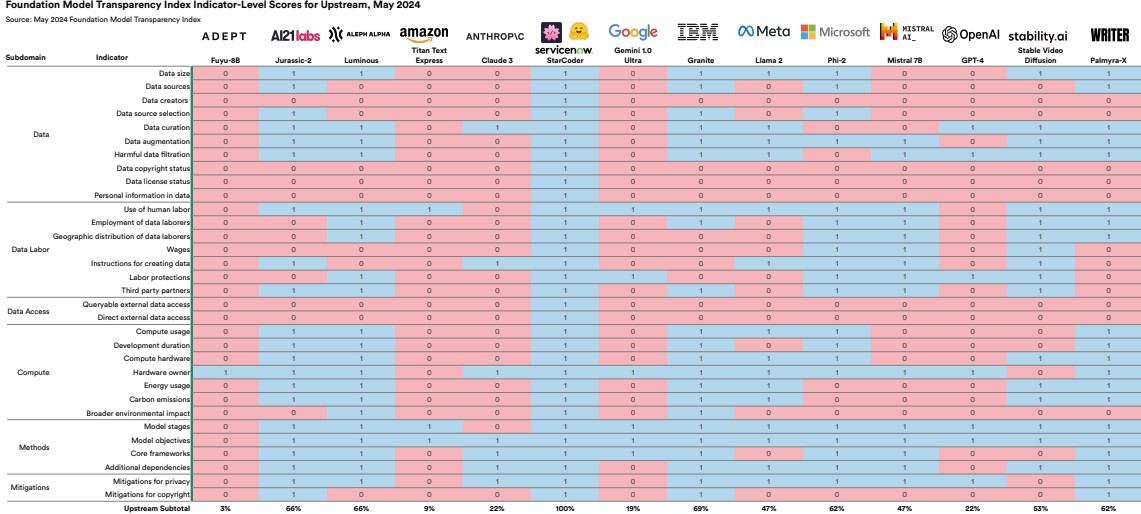

Figure 13: **Upstream Scores by Indicator.** The scores for each developer on each of the 32 upstream indicators.

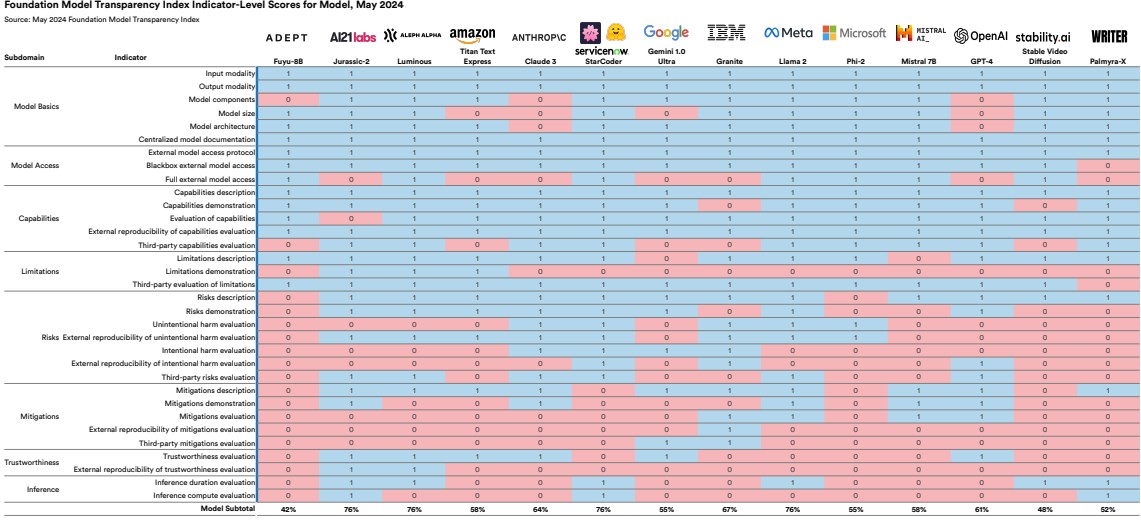

Figure 14: **Model Scores by Indicator.** The scores for each developer on each of the 33 model indicators.

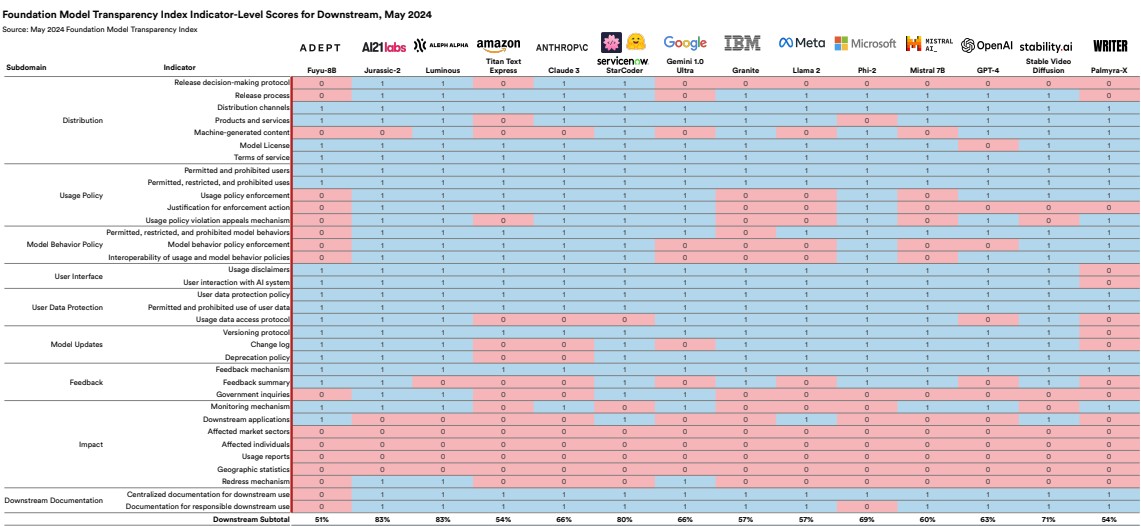

Figure 15: **Downstream Scores by Indicator.** The scores for each developer on each of the 35 downstream indicators.

