# OpenReview forum: "The 2024 Foundation Model Transparency Index"
_TMLR — Accepted by TMLR_

### Review · Reviewer_7TYK · 2024-07-27

**Summary Of Contributions:**

The paper presents a study on the transparency of major large language models (LLM). The paper builds on prior work on "Foundation Model Transparency Index" (FMTI) -- especially the choice of questions and previous estimation of the index using publicly available information, and presents an updated study where the authors engage in a discussion with LLM developers in order to get as much information as developers are willing to disclose, as well as resolving ambiguities regarding the questions.

The paper reports that significantly more information was disclosed with this new process, yet most developers tend to keep a lot of information private, in particular regarding data used for (pre)training.

**Audience:**

Yes

**Broader Impact Concerns:**

no concern per se, the paper contains a fairly exhaustive discussion on limitations

**Claims And Evidence:**

Yes

**Requested Changes:**

no changes requested, I think the paper satisfies TMLR's acceptance conditions

**Strengths And Weaknesses:**

strengths:

* I think this is a great initiative, it requires a lot of work, the report is serious and well-documented

* at least two results seem interesting to me:

1- most developers disclose more information when explicitly asked for it. This likely means that some information is not intentionally kept private

2- this effort allows to formalize which areas are systematically kept private. I am not sure we learn much here (e.g., it was pretty clear that developers do not want to say too much about the data they use), but this effort confirms that developers do not give precise information even when explicitly asked for it

weaknesses:

* the paper is not really standalone. There is a difficult trade-off because the paper heavily builds on the previous paper on the v1.0 report which is 115 pages long and difficult to summarize.  Figure 8 was useful to better understand the nature of the index. Somehow I missed the reference in section 3.1 in a first read. Probably because to me the need to better understand the questions came up when reading the results and I forgot about this reference. I don’t have any clear solution to that, apart from putting Figure 8 within the full body of the paper. I’m not sure it’s the best, but I think that would have helped me.

* the paper does not address "why" developers are shy to answer some of these questions, which feels like a missed opportunity. I think understanding why some information is kept private is at least as important as whether it is kept private

* probably my main comment/criticism:

1- the paper (and the previous one) does not seem to address the question of how much transparency should be expected from developers and what is legitimate to keep private. For instance, not disclosing the number of parameters of a model does not seem to be a big deal to me as far as the end users is concerned, but of course it doesn't help reproducibility. But many of these developers are for-profit companies, and it is not part of their mission to help reproducibility.

In the end, it is unclear to me how to interpret the score -- does "58" mean that observers do not have "enough" information on the model (e.g., to study the risks and impact), or it simply means that some information is legitimately kept proprietary.

2- some questions seem very difficult to answer to me and are more research questions. For instance, "who uses the model" and "how they use the model" seems quite difficult to answer accurately for open source models, and "who is impacted" also seems to be quite difficult as it is likely changing substantially every month and depending a lot on where we are looking into in the world.


* out of the scope of the current paper (and thus this review), but I am also not sure about the phrasing of some questions. For instance, asking about "wages" (wages of third-party data workers) and giving the point for the answer "larger than local minimum wage" is far from giving much information about worker exploitation -- e.g., this tells nothing about the working conditions or the type of working contract, which may be more meaningful (for instance, if the workers are full time employees rather than gig workers, this would imply a wage larger than local minimum).

---

> ### Author Response · Authors · 2024-10-14
> **Response to Reviewer 7TYK**
>
> We thank the reviewer for their extensive and supportive review.
>
> To respond to the issues raised:
>
> > the paper is not really standalone.
>
> To confirm, our understanding is the core challenge is that the previous paper is where the underlying indicators are defined, so understanding the present paper without understanding the indicators is challenging. This makes sense and we would make two changes to address this:
> (i) Moving Figure 8 into the main text as you suggest
> (ii) Providing a succinct description of the top-level domains and a few key subdomains in the main text with examples of a couple specific indicators.
>
> In doing so, we believe this will be a clear improvement: we will only add ~1.5 pages to the main text, but significantly help readers build intuition on the core criteria we use to score companies.
>
> > I think understanding why some information is kept private is at least as important as whether it is kept private
>
> We fully agree and, in fact, we did try to have companies articulate why they did not make information public. While a number of reasons are cited (e.g. competitive interest for information on building models, security/safety for information on model risks, privacy/contract limits for information on downstream usage, lack of infrastructure to measure in several cases), we ultimately did not discuss this too extensively because it felt fairly idiosyncratic, but we would be happy to add discussion if this seems valuable.
>
> One of the follow-ups we are considering is either (i) having companies justify why information is not public more formally in the 2025 FMTI or (ii) conducting a survey for companies to explain this.
>
> > does not seem to address the question of how much transparency should be expected from developers and what is legitimate to keep private. For instance, not disclosing the number of parameters of a model does not seem to be a big deal to me as far as the end users is concerned, but of course it doesn't help reproducibility. But many of these developers are for-profit companies, and it is not part of their mission to help reproducibility.
>
> We agree that the ultimate normative question of what the minimum standard for transparency *should* be is very important, and in fact one where our work is directly informing parallel efforts by governments regulating on foundation models (e.g. the EU AI Act, the US Foundation Model Transparency Act). With that said, our objective in this work was to first do the descriptive work:
> (i) What do companies disclose?
> (ii) Where can the Index itself, from 2023 to 2024, reveal that certain subdomains/indicators are malleable (i.e. in a span of 6 months, we can see a transparency improvement) vs. what is more stubborn to change?
> (iii) Where is there heterogeneity across companies?
>
> Alongside understanding the legitimate countervailing interests for transparency, we see our work as establishing the baseline. With our now improved empirical understanding, we can now address the normative question of what should be transparent more clearly (and world governments are using the 2024 report in precisely this way, because the answer to this question should make use of the expertise of many stakeholders beyond ML researchers).
>
>
> Please let us know if our responses address your core questions and if there is anything we could do that would improve the paper or clarifies matters. Thank you once again for your thorough review, we are very appreciative.

---

### Review · Reviewer_WyGb · 2024-09-02

**Summary Of Contributions:**

The paper provides insights into v1.1 of the "Foundation Model Transparency Index", which is a novel index to measure the transparency of foundation models released by leading developers.

The paper seems to be a technical report that highlights how the model was developed and the insights that the authors learned from using their model to score 14 companies.

**Audience:**

Yes

**Broader Impact Concerns:**

The goal of the paper is to address the broader impacts concern of transparency in foundation models.

If I had any nitpicky concern, it would be that the tool, if not comprehensive enough, could be used by companies to justify that they are doing things right and ethically when they are not. However, given that the model of the paper/the foundation model transparency index seems comprehensive and well thought-out, this seems like a very secondary worry. The improved visibility on how transparent companies are about their model largely negates any potential minor impact.

**Claims And Evidence:**

Yes

**Requested Changes:**

N/A

**Strengths And Weaknesses:**

Strengths:
- The problem that the authors studied is extremely important, practical, and has the potential to be extremely impactful.
- It is very clear that the authors went through a serious, comprehensive process of indicator selections. The authors have also access to 14 industry collaborators that they have been closely working with on refining and releasing their final indices.
- The indicators are clearly divided into downstream, upstream, and model-level indicators, which allows the authors to draw clear lessons as to where foundation model developers fail in terms of transparency. To me, the most exciting insight of the paper was how the authors' work highlights and quantifies how, despite significant progresses in terms of transparency, many foundation models still largely fail at being transparent when it comes to data collection, compute, and labor.
- The authors' has in fact already been impactful and may have helped/pushed companies to provide more transparent models: "the 8 developers evaluated in both October 2023 and May 2024 showed marked improvement,or 19 points on average (...) Developers that were assessed only in v1.1 performed slightly worse than those assessed in both v1.0 and v1.1".

Weaknesses:
- It might be useful to extend the peer review process to more reviewers. I am assuming that both reviewers are authors of the paper (let me know if wrong); even while the reviews were conducted independently, it is possible that the reviewers may have similar biases; involving more, outside reviewers may further confirm the validity of the study.
- I am not sure if this is really a weakness, but the paper itself does not make a technical/fundamental contribution to ML. Rather, the paper is more of a technical report on the actual technical/fundamental contribution, which the reviewers sadly cannot access directly without breaking the double-blindness of the process. I am actually wondering if this is a good fit for TMLR, due to the above. (Note that I would not want to reject the paper based solely on this comment, I am just thinking this should be a point of discussion).
I am also actually wondering if this is a missed opportunity by the authors to go for the smaller TMLR community when I believe this is the kind of work that could attract a large audience from non-ML expert/could be submitted to a higher reach publication where this index would be more publicized and have more impact.

Overall, I would support the paper being accepted if the rest of the review team believes this is a good fit for TMLR. This is a very nice, refreshing, impactful and transformative contribution.

---

> ### Author Response · Authors · 2024-10-14
> **Response to Reviewer WyGb**
>
> We thank the reviewer for their extensive and supportive review.
>
> To respond to the issues raised:
>
> > More scorers/reviewers of scores, including individuals independent from the FMTI team
>
> First, we want to emphasize that every score is reviewed and rebutted by representatives of the associated company, which significantly stress tests the scores. Beyond this, we do agree that have additional scorers that are more external would be good: we will consider implementing this for the 2025 FMTI, though it is a bit awkward to do at this point for the 2024 FMTI since the results are public and widely distributed. Regardless, we fully agree with the spirit of the suggestion and will think about incorporating it moving forward.
>
> > Venue fit
>
> We agree the paper is a bit different from papers generally published in ML venues (or TMLR specifically, though of course the venue is young and we hope to contribute to the breadth of papers published within it). With that said, we do feel papers of this kind are important to publish in top ML venues as ML becomes more and more important in society.

---

### Review · Reviewer_N9wk · 2024-09-30

**Summary Of Contributions:**

The paper advances the Foundation Model Transparency Index (FMTI) v1.0 presented in October 2023 to measure the transparency of leading foundation model developers by requesting developers to report the relevant information for each indicator. The analysis is based on 14 reports. The FMTI v1.1 results show room improvements in transparency over half a year.

**Audience:**

Yes

**Claims And Evidence:**

Yes

**Requested Changes:**

The requested change is therefore to present for publication a unique manuscript presenting FMTI v1.0 and v1.1 explaining all the differencing but having is as a primary contribution for the work. This would also avoid confusion to recognize the improvement of the paper, stating it as a unique one. Obviously, the authors should clearly state in the cover letter that the work already appeared online on a public access repository and that it already received concerns, at least in terms of the number of citations.

**Strengths And Weaknesses:**

The paper is well-written and structured and faces an ambitious and crucial challenge. The main issue is that the manuscript is very incremental w.r.t. FMTI v1.0 and largely based on that. Indeed, a reader cannot fully understand it without having read the other paper. This would not have been an issue if the FMTI v1.0 paper had been previously peer-reviewed and published. However, accepting for publication the FMTI v1.1 version means accepting, as a consequence, all the content of the paper presenting FMTI v1.0.

---

> ### Author Response · Authors · 2024-10-14
> **Response to Reviewer N9wk**
>
> We thank the reviewer for taking the time to review our paper and their positive response.
>
> We fully agree that the paper depends on the 2023 FMTI paper, but we highlight two key points:
> 1. We believe the methodological changes (i.e. getting companies to produce transparency reports) and the findings (i.e. how the status quo changed, the overall increase in transparency) are both sufficiently substantive and appropriate to publish. In parallel, we highlight that the 2023 FMTI paper is also under review.
> 2. We highlight that the paper cited in the background (Raji and Buolamwini, 2019) is a very similar paper, i.e. a follow-up study on the well known paper of Gebru and Buolamwini (2018) on the GenderShades audit of face recognition systems. Overall, we believe publishing a follow-up study, especially given the methodological improvements and key new findings, would make sense at TMLR.
>
> With this in mind, we will make changes that better articulate the relevant dependence on FMTI v1.0, which primarily is just the definition of the indicators (e.g. moving Figure 8 into the main text rather than the Appendix and providing more description of the specific domains and indicators).

---

### Decision · Action_Editor_S3MF · 2024-12-06

**Recommendation:** Accept with minor revision

**Comment:**

Reviewers WyGb and 7TYK gave thorough reviews and opine that the paper meets TMLR's criteria. However, the reviewers do acknowledge that this is not a "standard" TMLR paper and that it builds heavily on a previous report.

Reviewer N9wk initially opined that the paper met TMLR's criteria (while giving a really lightweight review), but then after reading the thorough reviews of reviewers WyGb and 7TYK, flipped their opinion.

Aggregating all the information from the reviewers and based on my own more brief reading of the paper, I think this paper makes contributions that are valuable to the community. While the writing should be improved (see last comment below), I think the claims are appropriate for the content and the community will be interested in this version 1.1 of the transparency index.

I request the authors to please address the following in their revision:

- Include your response to the reviewer's comment of "More scorers/reviewers of scores, including individuals independent from the FMTI team" as this is a natural question that a reader may also have.

- "I think understanding why some information is kept private is at least as important as whether it is kept private" Please add information on this as well, as you described in your response.

- Please add the details promised in response to reviewer comment "the paper is not really standalone." This will help make the paper more understandable. In case it helps, you have the flexibility that the published version need not preserve author anonymity.

**Audience:**

Yes

**Claims And Evidence:**

Yes